# The Role of Cognitive Abilities in Project-Based Teaching: A Mixed Methods Study

**DOI:** 10.3390/bs15030299

**Published:** 2025-03-03

**Authors:** Li Wang, Chunli Zhang

**Affiliations:** Faculty of Education, Beijing Normal University, Beijing 100875, China; 11132023048@bnu.edu.cn

**Keywords:** project-based teaching, cognitive abilities, spatial ability, primary school teacher, mixed methods study

## Abstract

Cognitive abilities are foundational to complex tasks, which may be also important in complex project-based teaching. However, the role of teachers’ cognitive abilities in project-based teaching is still unknown. Therefore, this study aimed to explore the relationship between teachers’ cognitive abilities and project-based teaching using a mixed methods design. In Study 1, a quantitative regression analysis was conducted with 62 primary school teachers. They completed the project-based teaching questionnaire and performed four cognitive tasks: remote association (creativity), object detail memory (object detail processing ability), paper folding (spatial ability), and sentence comprehension (verbal ability). Regression analysis revealed that spatial ability significantly predicted a teacher’s project-based teaching ability, even after controlling for age, gender, teaching experience, and project-based teaching experience. In Study 2, a qualitative exploratory case study was employed to examine how spatial ability manifests in two teachers’ project-based teaching plans. The teacher with higher spatial ability used schemata, abstract concepts, a better overall plan, and a deeper understanding of mathematics than teachers with lower spatial ability. This study indicated that a teacher’s spatial ability is closely associated with their project-based teaching, and it provides a new perspective for teachers to cultivate project-based teaching.

## 1. Introduction

Project-based learning and teaching are globally popular in education. For example, the project-based learning Works at the Buck Institute for Education, USA, has been providing professional development in science project-based learning to teachers and leaders at both the school and district levels since 1999 ([33]). Finland’s new National Curriculum for 2016 emphasizes interdisciplinary and integrated education to dismantle subject barriers and prepare students for future needs through project-based learning ([7]). According to the “Compulsory Education Curriculum Plan and Curriculum Standards (2022)” of the [44] ([44]), two basic principles for the compulsory education curriculum were implemented: one was to strengthen curriculum integration and focus on curriculum correlations and the other was to transform the educational approach to highlight practice, advocating for learning through doing, using, and creating.

As an essential part of project-based learning, project-based teaching must be given due consideration. Previous studies explored contextual and affective factors of project-based teaching (e.g., school policies and teacher motivation; [20]; [37]). However, cognitive abilities are foundational to complex tasks ([28]; [45]; [60]). As a complex teaching task, project-based teaching may also require cognitive abilities. Currently, no studies have explored the role of cognitive abilities in project-based teaching. Therefore, this study aims to investigate how teachers’ cognitive abilities are associated with project-based teaching.

### 1.1. Project-Based Teaching

Project-based teaching is essentially another aspect of project-based learning but for different subjects. Generally, project-based learning is a learning process for students, while project-based teaching refers to a process through which teachers design and guide students in implementing project-based learning in class ([21]). Project-based teaching has no single, precise definition; its advocates generally agree on certain basic characteristics. For example, [23] ([23]) reviewed 48 articles from 2004 to 2014 and proposed five main features to define project-based science and technology teaching and learning: there is an authentic scientific question; the students develop a final product; the students are engaged in investigations or design activities; there is collaboration among students, teachers, and others in the community; and learning technologies are used. [21] ([21]) proposed basic characteristics of project-based teaching, including giving students opportunities to study a challenging problem, engaging in sustained inquiry, finding answers to authentic questions, helping choose the project, reflecting on the process, critiquing and revising the work, and creating a public product. More importantly, the Buck Institute for Education proposed seven more practical gold standards for project-based teaching, including designing and planning, alignment to the standard, building the culture, active managing, scaffolding student learning, assessing student learning, and engaging and coaching ([38]).

Compared with the considerable body of literature on project-based learning ([32]; [34]; [63]), relatively little attention has been paid to project-based teaching. Solely paying attention to how teachers guide students to carry out project-based learning is project-based teaching. Some studies about teachers’ learning through project-based learning are essentially project-based learning rather than project-based teaching (e.g., [30]; [43]). For example, [30] ([30]) examined how project-based learning can be used as a professional development approach for teachers, emphasizing the importance of engaging in authentic, inquiry-based projects to enhance their instructional practice.

Previous studies on actual project-based teaching explored a wide range of influencing factors and aspects related to its implementation and effectiveness. For instance, research has examined how gender, implementation experience, and school management impact project-based teaching ([20]). Attitudes towards project-based learning have also been a focus, with findings indicating that positive attitudes can significantly enhance student engagement and learning outcomes ([22]). Additionally, multimedia resources are a crucial factor in supporting project-based teaching, as they can provide students with more interactive and immersive learning experiences ([46]). Teacher motivation has been identified as another key factor, with motivated teachers being more likely to effectively implement project-based teaching and support student learning ([37]). Professional identity has been highlighted as an important aspect, with teachers with a strong professional identity being better equipped to navigate the challenges of project-based teaching ([56]). Despite these extensive investigations, there remains a gap in the literature regarding the role of teachers’ cognitive abilities in project-based teaching.

This study is situated in China, where project-based learning and teaching are in a stage of rapid development. Before 2020, there was not much research on project-based teaching in China, and the total number of Chinese core journal papers published was about 50. Nearly 140 Chinese core journal papers have been published between 2020 and 2025 with the titles of “project-based teaching”. These studies probe into how project-based teaching works in different specific contexts, such as the “Construction of interdisciplinary project-based teaching model in Geography” ([52]), “Project-based teaching of VB program from the perspective of problem solving” ([31]), and so on. Benefiting from the new national policy of the “Compulsory Education Curriculum Plan and Curriculum Standards (2022)”, project-based learning and teaching has become a new national way of compulsory education. With many teachers needing to carry out project-based teaching, ways to better cultivate their project-based teaching are urgent. Despite extensive studies conducted in China and other countries, there remains a gap in the literature regarding the role of teachers’ cognitive abilities in project-based teaching.

### 1.2. Cognitive Ability Model

Cognitive ability refers to any ability that concerns the processing of mental information ([10]). Such cognitive abilities are foundational to complex tasks. For example, studies on problem solving highlight that spatial and verbal abilities are critical for representing abstract relationships and coordinating multi-step processes ([28]; [45]; [60]). As a complex teaching task, project-based teaching may also require cognitive abilities. For example, teachers with higher spatial ability demonstrate superior curriculum integration in STEM fields ([58]). Meanwhile, teachers with stronger verbal abilities excel in scaffolding student discussions ([24]). Crucially, cognitive abilities are not merely “static traits” but malleable skills that can be enhanced through targeted training ([25]; [49]; [58]). By examining teachers’ cognition, this study seeks to uncover actionable pathways for professional development.

Cognitive ability is not unidimensional; a massive amount of empirical evidence from factor analysis suggests the use of multidimensional models of cognitive ability tests (e.g., [3]; [10]; [45]; [51]; [53]). Researchers have proposed many different multidimensional models. For example, [53] ([53]) proposed two general factors: abilities that are acquired (crystallized intelligence or gc) and abilities that reflect natural potential and the biological integrity of the cerebral cortex (fluid intelligence or gf).

This study mainly focused on the latest object–spatial–verbal cognitive model ([3]). It has been suggested that object detail, spatial, and verbal processing may represent three different informational processing systems. Although they are synergistic, there are some structural and functional separations. For example, in complex spatial tasks, keeping too many images or object details leads to an overload of visual working memory and a reduction in the efficiency of spatial processing ([35]). Similarly, spatial processing and verbal processing may interfere with each other. For example, when verbal and spatial processing are associated with the same hemisphere (i.e., both right or both left), a significant decrease in verbal comprehension performance is found. However, the score increases if they are associated with different hemispheres ([47]). The cortical surface better predicts spatial ability than thickness, whereas neither factor predicts language ability ([62]). The object–spatial–verbal cognitive model clearly distinguishes the function of three types of cognitive systems from the basis of brain regions ([3]). For example, spatial ability is associated with the dorsal stream ([12]), verbal ability is dependent on the left temporal lobe ([8]; [29]), and object processing is dependent on the ventral stream ([2]; [9]). It justified testing each ability’s unique contribution to project-based teaching rather than aggregating them into a single “general intelligence” metric. This helps to accurately identify key cognitive components in project-based teaching.

In addition, creativity may also be closely related to project-based teaching. For example, after a semester of project-based learning, college students’ creative thinking was found to significantly improve ([17]). Therefore, creativity was also added to the assessed cognitive abilities in this study.

### 1.3. Current Study

Based on the seven gold standards of project-based teaching ([38]), the current study investigated project-based teaching following the five core elements, which are analyzing textbook knowledge to align with standards, designing or planning projects, implementing projects in class, assessing all aspects of project-based learning, and managing or controlling active in-class learning. There are two main differences between the seven gold standards. First, no “build the culture” standard is included due to implementation challenges. This standard asks teachers to explicitly and implicitly promote student independence and growth, open-ended inquiry, team spirit, and attention to quality ([38]). This shows that the standard itself has a character of implicitly. Moreover, the different school cultural backgrounds, student diversity, and uneven teaching resource distribution can make it challenging to implement cultural-related standards. For example, a qualitative study showed that elementary teachers may not be teaching in a culturally responsive way due to a lack of classroom models ([26]). Second, we combined the “scaffolding student learning” standard with the “engage and coach” standard into a project implementation element. The “scaffolding student learning” emphasizes providing a variety of strategies to support students, while “engage and coach” focuses on the dynamic interaction of students and teachers in the process ([38]). [59]’s ([59]) “zone of proximal development” theory pointed out that learning should be realized through scaffolding and interaction under the guidance of teachers. With this combination, teachers can both provide instrumental support and dynamically adjust instruction through observation and feedback. In practice, scaffolding and coaching often go hand in hand. Teachers who scaffold student learning already engage and coach students by providing the necessary support and guidance.

This study aimed to explore how teachers’ cognitive abilities are associated with project-based teaching and to quantitatively identify cognitive correlates of project-based teaching. The first research question is as follows: which cognitive abilities predict project-based teaching? Based on an object–spatial–verbal cognitive model ([3]) and the importance of creativity (e.g., [17]), we designed Study 1 to measure cognitive abilities from four aspects: object ability, spatial ability, verbal ability, and creative ability. The relationship between these cognitive abilities and the scores of the project-based teaching questionnaire was explored in Study 1. Because these cognitive abilities are relatively independent in the cognitive model ([3]), we hypothesized that only one or two cognitive abilities may play a unique role in project-based teaching.

To quantitatively identify the cognitive correlations of project-based teaching, we posed the second research question: how does a key cognitive predictor specifically shape the design and implementation of project-based teaching? Furthermore, by the case study of contrasting high vs. low cognitive ability teachers’ project-based teaching plans, Study 2 could illuminate actionable strategies that align with the cognitive abilities, offering concrete recommendations for teacher education.

## 2. Study 1

Study 1 aimed to explore the relationship between cognitive abilities and project-based teaching. Four cognitive abilities were measured—object ability, spatial ability, verbal ability, and creative ability—and a project-based teaching questionnaire was conducted in this study. Object ability was measured using an object detail memory task ([6]), spatial ability was measured using a paper folding task ([16]), and verbal ability was measured using a sentence comprehension task ([13]). The creative ability was measured using a remote association task ([42]). The project-based teaching questionnaire was adopted from a related master’s thesis in China ([41]). Moreover, we controlled for important covariates to explore the unique role of cognition in this study, such as age, gender, teaching experience, and project-based teaching experience. For example, meta-analysis evidence suggests that males tend to outperform females in spatial tasks ([39]; [40]). Cognitive abilities (e.g., spatial ability and working memory) typically decline with age, particularly after early adulthood ([50]). While teaching experience and project-based teaching experience are directly associated with project-based teaching, we expect that only some cognitive abilities play a unique role in project-based teaching after controlling for age, gender, teaching experience, and project-based teaching experience.

### 2.1. Methods

#### 2.1.1. Participants

This study included 62 primary school teachers from 42 schools in Beijing and Hebei, China. These regions started project-based teaching earlier than most regions in China and are more representative. The average age of the teachers was 35.11 ± 5.98 years old, with a total of 8 males and 54 females. Their average teaching experience was 12.58 ± 6.54 years. These teachers were exposed to project-based teaching for approximately 2 years. Among subjects, there were 21 Chinese teachers, 16 math teachers, 12 science teachers, and 13 teachers of other subjects (including English, ethics and law, and art and sports). The grades currently taught were one through six. These teachers’ highest degree was a bachelor’s or master’s degree. These teachers were measured by voluntarily enrolling in a teacher training program. Informed consent for participation was obtained from all subjects involved in this study.

#### 2.1.2. Tasks

The project-based teaching questionnaire was based on five core elements of project-based teaching, and a related master’s thesis in China ([41]). The questionnaire consisted of five abilities in project-based teaching and designed a questionnaire containing five multiple-choice questions. The first question is the analysis of textbook knowledge. This question further contained three options, and the teachers chose the option to indicate that they had this ability. Similarly, the other four core elements (design or plan project, implementation project, assess projects, manage or control active in class) are designed as four questions. They selected the corresponding option and were awarded 1 point. The Z-scores of the five questions were added to the total score. Please see all the questions in Appendix A. The original version of this questionnaire was used on 213 Chinese primary school teachers, and it was found that its reliability (Cronbach’s α coefficient is 0.97) and validity (the correlation of total score and each question larger than 0.40) were good ([41]).

Study 1 included four cognitive tasks, and the details of these tasks are as follows (Figure 1).

Remote association task. This test mainly examined the remote association ability of Chinese words, which is an aspect of creative thinking ([42]). Three Chinese cue characters were presented at the top of a screen, and the participants needed to find a target character that could be combined with the cue words to form legitimate Chinese two-character words. For example, the cue characters were “板 (Board), 洞 (Hole), 色 (Color)”, and the target character was “黑 (Black)”, which form the two-character words “黑板 (Black board), 黑洞 (Black hole), 黑色 (Black color)”, respectively. The task consisted of 20 questions with a time limit of 3 min, and the task automatically ended when the time had expired.

Paper folding. This task estimated spatial ability, especially spatial visualization ([16]). This corresponds to the spatial dimension of the object–spatial–verbal model, which involves the ability to understand and navigate spatial configurations. A piece of paper appeared on the screen with directional folds indicated by arrows and punched holes. The participants were asked to look at images of unfolded paper containing holes and select the correct corresponding image from the five options. The task consisted of 20 questions with a time limit of 3 min, and the task automatically ended when the time had expired.

Sentence comprehension. This test measured verbal ability ([13]). This aligns with the verbal dimension of the object–spatial–verbal model, which focuses on the comprehension and production of linguistic information. A sentence was presented on the screen, and the participants needed to find a suitable word to finish the sentence. For example, “The root cause lies in the of critical thinking and rationality. In the progress and development of society, the spirit of science is a powerful tool to eliminate various and break through all sorts of superstitions (A). A. Absence, Foolishness; B. Scarcity, Evil; C. Weakness, Ignorance; D. Failure, Misunderstanding” The test consisted of 20 questions with a time limit of 3 min, and the task automatically ended when the time had expired.

Object detail memory. The test measured the ability to remember details of objects ([6]). This aligns with the object processing dimension of the object–spatial–verbal model, which focuses on the detailed and accurate representation of visual stimuli. The task included two phases: learning and testing. In the learning phase, the participants memorized 40 pictures of normal objects one by one, with each picture presented for 3 s at an interval of 1 s. In the testing phase, there were 20 questions, in which the participants needed to choose one of two pictures that they had seen during the learning phase. There was no time limit.

#### 2.1.3. Data Analyses

The data were sorted and analyzed using SPSS 24.0. First, the descriptive statistics of cognitive abilities and project-based teaching were determined. Then, a correlation of cognitive ability’s role in project-based teaching was conducted to reveal its relationship. Finally, a regression analysis of control important covariates (i.e., age, gender, teaching experience, and project-based teaching experience) was conducted to reveal the unique role of cognitive abilities in project-based teaching.

### 2.2. Results

#### 2.2.1. Descriptive Statistics Results

The descriptive statistics results shown in Table 1 revealed that Cronbach’s α reliability scores for all the cognitive tasks and the project-based teaching questionnaire were larger than 0.7, indicating good reliability. The remote association task was relatively difficult, resulting in lower scores than for the other cognitive tests, and the distribution was positively skewed (skewness: 2.21, kurtosis: 8.14).

#### 2.2.2. The Project-Based Teaching Investigation Results

For the project-based teaching questionnaires, the scores for each option are shown in Figure 2. First, most of the primary school teachers had knowledge analysis abilities regarding curriculum structure and knowledge (66.12%) and scope of application knowledge (59.68%). These two skills are associated with traditional teaching, but when asked about the association with knowledge and project-based learning, many teachers did not consider themselves to have this skill (32.26%). This shows that teachers still lack the knowledge analysis ability needed for project-based teaching.

Second, most of them could design appropriate production forms (53.22%) and set appropriate project-based learning goals (50.00%). This shows that teachers’ understanding of project-based learning may start from the goal and be marked by the result, but they are still confused about the design of specific content.

Third, the implementation abilities were mainly organizing students into appropriate groups (77.41%) and using various devices (67.74%). Since teachers know the students in their own class well, they generally believe that grouping is controllable. But, there is a lack of specific guidance or scaffolding for students to carry out project-based learning activities.

Fourth, the main assessment abilities lay in the use of processing assessments (67.74%) and multiple subjects of assessment (58.06%). Although there is a lack of use of multiple-form tools, the overall situation is not bad (45.16%). This shows that teachers have a good grasp of the assessment of project-based learning in class.

Fifth, most primary school teachers had all classroom control abilities, including finding problems that hinder activities (74.19%), organizing and managing students well (77.41%), and adjusting the tasks and progress in time (59.67%). This shows that the teachers in the questionnaire have a strong ability to control their own classes, which may also be one of the characteristics of Chinese teachers.

In summary, the current results indicated that primary school teachers already have most project-based teaching skills.

#### 2.2.3. The Correlation Analysis Results

The correlation between cognitive abilities and project-based teaching is shown in Table 2. The remote association task significantly correlated with object detail memory, r = 0.34 and *p* = 0.007. The absence of significant correlations between other cognitive abilities illustrated the separation of the object–spatial–verbal cognitive model. All project-based teaching elements correlated with each other with total scores, r, from 0.36 to 0.90 and *p* < 0.01. Thus, the project-based teaching questionnaire was well designed and had good internal consistency. Only the cognitive ability of paper folding significantly correlated with all the project-based teaching elements, with r from 0.26 to 0.38 and *p* < 0.05. This showed the important role of spatial ability in primary school teachers’ project-based teaching elements.

#### 2.2.4. The Regression Analysis Results

A regression analysis of cognitive abilities on project-based teaching elements after controlling for age, gender, teaching experience, and project-based teaching experience is shown in Table 3. Statistical significance was set at a Bonferroni-corrected *p*-value of 0.008 (0.05/6 because there were six models). Paper folding significantly predicted the total score of project-based teaching, with β = 0.39 and *p* = 0.006; knowledge analysis ability, with β = 0.39 and *p* = 0.006; and implementation ability, with β = 0.40 and *p* = 0.005. This further showed the important role of spatial ability in project-based teaching. Moreover, the project-based teaching experience significantly predicted the design ability, with β = 0.42 and *p* = 0.003. This shows that the design of project-based teaching is closely related to prior experience.

## 3. Study 2

In Study 1, we found that among all the cognitive abilities, only spatial ability was associated with project-based teaching. However, relying on self-reported measures for project-based teaching may introduce bias. For example, the Dunning–Kruger effect, a cognitive bias wherein individuals with low ability in a domain overestimate their competence, can significantly impact self-report questionnaires ([36]). Moreover, quantitative regression (Study 1) cannot reveal why spatial ability matters. Therefore, in Study 2, we chose project-based teaching plans completed by two teachers with different spatial abilities to further demonstrate the relationship between spatial ability and project-based teaching. This case analysis involves an in-depth examination of spatial ability’s role in project-based teaching, providing a more objective and detailed understanding to avoid self-report biases. Although the sample size of Study 2 is small, it is still valuable. On the one hand, the design orientation of Study 2 is an exploratory case study. The core objective is not to generalize statistics but to explore the potential mechanisms of the application of cognitive ability in educational practice. Such a case study is of great value in the early stage of theory construction or method validation ([18]). On the other hand, Study 2 will have a unique scientific contribution to the current study. Through two typical cases, we could verify the operation process and sensitivity of the spatial ability in project-based teaching, which laid the foundation for the follow-up large-sample research. And, the consistency of the qualitative data with the quantitative results enhances the credibility of the overall conclusion ([14]).

### 3.1. Methods

#### 3.1.1. Participants

After the data were collected in Study 1, the teachers began to develop project-based teaching plans, but only less than 10 teachers completed their final plans. Therefore, Study 2 selected two teachers from Study 1 participants with different spatial abilities and completed the final plans. Teacher A had a good spatial ability (paper folding score of 10), and teacher B had a lower spatial ability (paper folding score of 3), but there were no differences in the other aspects; see details in Table 4. For example, they were both mathematics teachers from Beijing, China. There were no differences in the other cognitive abilities, including remote association tasks, sentence comprehension, and object detail memory.

#### 3.1.2. Data Collection and Analyses

The two primary school teachers summarized their project-based teaching plans and shared the documents with the authors of this study. These documents were stored in anonymized digital formats. They were asked to summarize their project-based teaching plans using a fixed template. This template was designed according to the five core elements of project-based teaching given in this study. As the plans were presented in text form, the classroom control ability was not analyzed in this study. Therefore, this template includes four parts: textbook knowledge analysis, project-based learning design, implementation procedures, and assessment plans (see details in Appendix B).

This study mainly explores the embodiment of spatial ability in project-based teaching plans. Therefore, the analysis process mainly extracts information related to spatial ability, such as the use of schemata ([4]; [5]; [28]), abstract concepts ([27]), overall plan ([19]), and more. Differences were found through a qualitative comparison of the two plans.

### 3.2. Results

#### 3.2.1. Teacher A: Higher Spatial Ability

Teacher A’s project-based teaching plan was “park route planning”, which required students to provide personalized tour route services to address the different needs of tourists and effectively improve their satisfaction.

In the textbook knowledge analysis, she drew a knowledge logical structure diagram. In the diagram, the relationships between knowledge points were expressed through schemata. Schematic representations (drawn pictures that can represent quantitative relationships between mathematical elements) are related to the accuracy of word problem solving, as well as spatial ability ([4]; [5]; [28]). For teachers, the use of a schema in a knowledge analysis may explain the association between project-based teaching and spatial ability.

In the project-based learning design, teacher A designed a more abstract and general phase of the project. For example, the phases were as follows:


*“1: Project release–clarification of issues; 2: Project exploration–problem solving; and 3: Project optimization–problem reflection.”*


Perhaps for the design of project-based learning, more abstract and general concepts are needed to facilitate its transfer to the design of other projects. At the same time, spatial ability is also a more general, more abstract form of picture expression ([27]). The role of spatial ability is reflected in the abstract of the design phase of project-based teaching rather than the concrete design content. This may also be the reason why the relationship between spatial ability and project-based teaching design ability was not found in Study 1. The evaluation of design ability in Study 1 did not focus on abstractness.

In the implementation procedure, teacher A spent fewer class hours on the project introduction (1 class) than on the project design (2 classes). This showed that teacher A had better planning for teaching implementation. This ability may be related to spatial ability because the latter is the ability to comprehend the overall abstract relationship, which contributes to overall planning ability ([19]).

For assessment plans, teacher A’s assessment was closer to the essence of mathematics. For example, the assessment was as follows:


*“Excellent: according to the needs of specific groups, the development of special tour routes are in line with the standard, and the route is scientific and reasonable. Good: according to the needs of specific groups of people, the design of special tour lines was implemented, but there is a slight gap with the standard. Qualified: able to design characteristic tour routes, but the design is not reasonable.”*


This shows the teacher’s understanding of the nature of mathematics, which in turn affects the depth of the assessment criteria. At the same time, the understanding of mathematics is closely related to spatial ability ([5]; [28]; [25]). This also explains why Study 1 did not find a relationship between spatial ability and project-based teaching assessment ability, possibly because this association is closely related to the specific subject.

#### 3.2.2. Teacher B: Lower Spatial Ability

Teacher B’s plan was “when art meets mathematics”, which required that to learn to appreciate Escher’s dense-tiled layout artwork, students need to use periodic changes and graphical transformations in mathematics to design and create their own dense-tiled layout artwork.

In the textbook knowledge analysis, teacher B used the pure text description method. For teachers, the use of a pure text description in a knowledge analysis is related to verbal ability. According to the interferences and trade-offs among abilities in the object–spatial–verbal cognitive model ([3]), the advantage of verbal ability may limit the use of spatial ability. Therefore, when teacher B shows a greater preference for verbal ability, his spatial ability may be weaker.

In the project-based learning design, teacher B’s phase was more concrete. For example, the phases were as follows:


*“1: Art appreciation; 2: Concept understanding; 3: Art processing; and 4: Creative design.”*


Compared with teacher A, his design of project-based learning focuses on a specific project. This concrete and specific thinking may not facilitate spatial ability.

In the implementation procedure, teacher B spent more class hours on the project introduction (two classes) than on the project design (one class). This shows that he does not have a good overall plan for the implementation of project-based learning. And, this also explains the lack of overall representation ability of spatial ability.

For assessment plans, teacher B’s assessment was not related to the essence of mathematics. For example, teacher B’s assessment was as follows:


*“Excellent: understood the design principle of dense-tiled layout, completed dense-tiled layout art work, which was beautifully designed. Share design ideas, creative processes, and challenges. Good: understood the design principle of dense- tiled layout, basically completed a dense-tiled layout art work, with a certain artistic processing. Share design ideas. Qualified: Understood the design principle of dense-tiled layout, and basically completed a dense-tiled layout art work.”*


Obviously, teacher B’s assessment was not closely related to mathematical ability but was based on art and sharing. This may be related to his lack of spatial ability.

## 4. Discussion

This study aimed to explore the associations between teachers’ cognitive abilities and project-based teaching. The results showed that primary school teachers’ creative, object, and verbal abilities cannot predict project-based teaching well, with only spatial ability significantly predicting project-based teaching, including knowledge analysis and implementation. These associations were further demonstrated in a qualitative study, where the two primary school teachers with different spatial abilities showed some differences in project-based teaching plans. These results contribute to understanding the relationship between teachers’ cognitive abilities and project-based teaching, especially spatial ability. Even though students’ cognitive abilities are related to project-based learning abilities ([54]), not enough attention has been paid to teachers’ cognitive abilities and project-based teaching in the past. This study provides a new perspective for teachers to cultivate project-based teaching.

### 4.1. The Situation of Project-Based Teaching

As can be seen from the questionnaire results based on the project-based teaching questionnaire, teachers are more equipped with traditional teaching abilities but lack the ability directly related to project-based learning. For example, Study 1 showed that most teachers could analyze the curriculum structure and scope of knowledge but could not combine this knowledge with project-based learning. Most teachers have difficulty implementing project-based teaching, such as guiding students from project plans and methods, implementing remedial teaching, or organizing student projects to display. This suggests that the teacher group we investigated has not yet mastered project-based teaching. These situations are consistent with other studies on the challenges of project-based teaching, including various aspects such as teachers’ implementation skills, managing time for project-based teaching, organizing projects, and others ([1]). And, Study 1 found that project-based teaching experiences have a significant role in project-based teaching design ability. This indicated that the relationship between spatial ability and project-based teaching design may be affected by project-based teaching experience. More careful studies are needed in the future.

Notably, the participants were primary school teachers exposed to project-based teaching for approximately two years in China. On the one hand, elementary school teachers often have basic knowledge compared to specialized secondary school teachers. Spatial ability may be a key in basic knowledge or elementary school project-based teaching. The effect of spatial ability on project-based teaching among teachers of other grades needs more research. On the other hand, the 2 years of experience with project-based teaching is significant, as it suggests that the teachers had sufficient time to develop and refine their practices within a project-based teaching framework. However, it also implies that these teachers were still adapting to and integrating project-based teaching into their teaching methods, which may influence the findings. Third, the participants in the current study were from China. Project-based teaching has not been popular in China for long, so most teachers lack extensive experience with project-based teaching. It is precisely because Chinese teachers need more relevant training that the current study is significant for project-based teaching training in China.

### 4.2. The Role of Spatial Ability in Project-Based Teaching

This study demonstrated the critical role of teachers’ spatial ability in project-based teaching. Individuals with higher spatial abilities tend to excel in tasks that require the manipulation and understanding of spatial relationships ([16]; [45]; [61]), which are essential components of problem solving in various fields ([4]; [5]; [28]). Regarding project-based teaching, individuals with strong spatial abilities are better equipped to analyze knowledge, visualize potential relationships, and effectively implement abstract project-producing strategies. The object–spatial–verbal cognitive model proposes interferences and trade-offs among abilities due to competition for limited executive resources ([3]). When spatial abilities are advantageous, object and verbal abilities may not be significant.

Spatial ability also manifests in various forms in teachers’ implementations of project-based teaching projects. First, using more schemata is a typical manifestation of spatial ability ([4]; [5]; [28]). Representing relationships between elements through pictures helps teachers think about concepts more deeply and abstractly. Second, the use of more abstract concepts is also a reflection of spatial ability. Spatial ability reflects an understanding of more general, abstract forms of pictorial expressions ([27]; [55]). Third, a better grasp of overall is also related to spatial ability. Indeed, spatial ability is related to abstract relationships and contributes to the overall planning ability ([19]). Thus, these three manifestations complement each other and can be used differently.

In the field of project-based teaching, previous studies on teachers’ project-based learning have explored the influencing factors of gender, implementation experience, and school management ([20]); professional experience and teaching direction ([48]); attitudes towards project-based learning ([22]); multimedia resources ([46]); teacher motivation ([37]); professional identity ([56]); and more. Unlike these studies, our study aimed to fill the gap in the literature regarding the role of teachers’ cognitive abilities in project-based teaching. It uncovered an actionable pathway for professional development from a cognitive perspective. In terms of cognitive studies, this study provides new evidence for the practical application of the object–spatial–verbal cognitive model. The results of spatial ability alone have a significant predictive effect, indirectly demonstrating the separation and conflict between object, spatial, and verbal abilities. Moreover, the current results demonstrate the particularity and importance of spatial ability in many cognitive abilities.

### 4.3. Implications

Because spatial ability among adults aged 19 to 65 years is amenable to enhancement through targeted training interventions ([49]), additional training focused on spatial ability should be incorporated into teacher professional development programs concerning project-based teaching. Specific suggestions are as follows.

First, teachers should be encouraged to use schemata in project-based teaching plans. Using more schemata is a typical manifestation of spatial ability ([4]; [5]; [28]). Representing relationships between elements through pictures helps teachers think about concepts more deeply and abstractly. When designing a project, teachers should pay attention to the integration and classification of the relevant information. After collecting a large amount of fragmentary information, they should classify it according to a certain logical relationship and internal order so that the schema can be constantly supplemented and a perfect schema system can be formed. Where necessary, we can even facilitate schema construction by providing related tools.

Second, teachers should be encouraged to use more abstract, overall, and core concepts in project-based teaching plans. The use of more abstract concepts is also a reflection of spatial ability. The nature of spatial ability is a more general, abstract form of pictorial expressions ([27]). It also contributes to the overall planning ability ([19]). Using more abstract and overall concepts in project-based teaching plans reflects the individual’s more profound understanding of project-based learning and teaching. After project-based teaching, teachers can reflect, summarize, and refine core ideas. Through repeated rounds of refining, teachers’ abstraction level of project-based teaching could be improved.

Third, an assessment system for teachers’ project-based teaching, including cognitive abilities, could facilitate the development of spatial ability. [11] ([11]) proposed instructional alignment, a process where the critical elements of instruction—learning objectives, assessment of learning, and anticipated knowledge, skills, or dispositions—interact and support learning outcomes. Therefore, to better promote the improvement of spatial ability, it is necessary to construct an assessment system first and use assessment to encourage improving project-based teaching.

### 4.4. Limitations and Future Suggestions

First, the small sample size poses limitations to the generalizability of the findings. With only 62 primary school teachers participating in Study 1, the quantitative results may not fully represent broader regions, schools, and so on. A smaller sample size can limit the statistical power of correlation and regression results, potentially leading to less reliable and less robust results. Researchers could divide the population into strata based on key characteristics (e.g., regions, age, teaching subjects) and use large-scale stratified sampling. More critically, the qualitative case study (Study 2) included only two teachers, which does not account for variations in teaching styles, educational contexts, or individual teacher characteristics. Small qualitative samples also risk premature theoretical saturation, where emergent themes may reflect idiosyncratic experiences rather than generalizable patterns ([15]). For example, the contrasting cases in Study 2 may overemphasize spatial ability’s role while neglecting other contextual factors (e.g., school resources, student levels) that influence project-based teaching. To address these limitations, in addition to expanding the sample size, future studies could use multiple data sources (e.g., interviews, observations, student feedback) to provide a more comprehensive understanding. Moreover, a coding scheme could be developed to help researchers systematically analyze a large amount of data.

Second, there are some limitations of the measurement tools. The measurement of project-based teaching was mainly based on self-reporting (Study 1) and text analysis (Study 2). The self-reported measures may introduce bias, such as the Dunning–Kruger effect, and so on ([36]). The text analysis may struggle with language characters, such as slang, jargon, or domain-specific language ([57]). Incorporating observations from real classroom interactions would have enriched the analysis and provided a more representative depiction of teaching practices. Perhaps based on new technologies, such as virtual reality and artificial intelligence, immersive and personalized project-based teaching ability measurement can be achieved in future studies. In the measurement of cognitive abilities, these are measures of domain-general abilities, but they do not integrate well with educational contexts. For example, remote association tasks may only capture linguistic creativity rather than nonverbal creativity (e.g., visualizing and constructing complex tasks) or pedagogical creativity (e.g., adapting projects to diverse learners). The sentence comprehension task does not account for critical instructional and communicative skills that are essential for guiding learners in a project-based environment. The object detail memory task also showed no meaningful association with teaching elements. This may be an important reason for the lack of associations between primary school teachers’ creative, object, and verbal abilities and project-based teaching. Future studies should use a variety of tasks that capture different aspects of cognitive abilities, such as mathematical creativity, speech intelligibility, educational object detail memory, and so on. Future studies also could combine cognitive tests with classroom observations and AI-driven analytics to develop interactive and implicit measurement tools. For example, mathematical teachers can demonstrate operations such as rotation, symmetry, and translation of graphs in class. AI platforms can record the reaction time of these operations in real time and automatically analyze the data related to spatial ability.

Finally, while our study suggests that incorporating additional training focused on spatial ability into teacher professional development programs could be beneficial, we recognize the limitations of our research. Future studies should aim to validate these findings with more extensive and diverse samples and explore the long-term impact of spatial ability on teaching practices. Furthermore, professional development programs should be designed to address the specific needs of teachers in different educational contexts.

## 5. Conclusions

Through a comprehensive approach involving both quantitative (Study 1) and qualitative (Study 2) analyses, this study revealed that spatial ability could significantly predict elementary school teachers’ project-based teaching ability. Specifically, Study 1 found that 62 primary school teachers’ spatial ability, rather than creativity, object detail processing ability, and verbal ability, significantly predicted teachers’ project-based teaching ability, even after controlling for age, gender, teaching experience, and project-based teaching experience. Study 2 of the case analysis found that the higher spatial ability teacher used more schemata, abstract concepts, a better overall plan, and a deeper understanding of mathematics in project-based teaching plans than teachers with lower spatial ability. This study not only highlighted the importance of spatial ability in project-based teaching but also uncovered an actionable pathway for professional development from a cognitive perspective. And, the mixed methods of qualitative and quantitative analysis enhance the credibility of the overall conclusion. This study has some limitations. For example, the sample size was small; the cognitive measurement tools did not well integrate with educational contexts, and so on. Future studies should aim to validate these findings with more extensive and diverse samples and explore the long-term impact of spatial ability on teaching practices.

## Figures and Tables

**Figure 1 behavsci-15-00299-f001:**
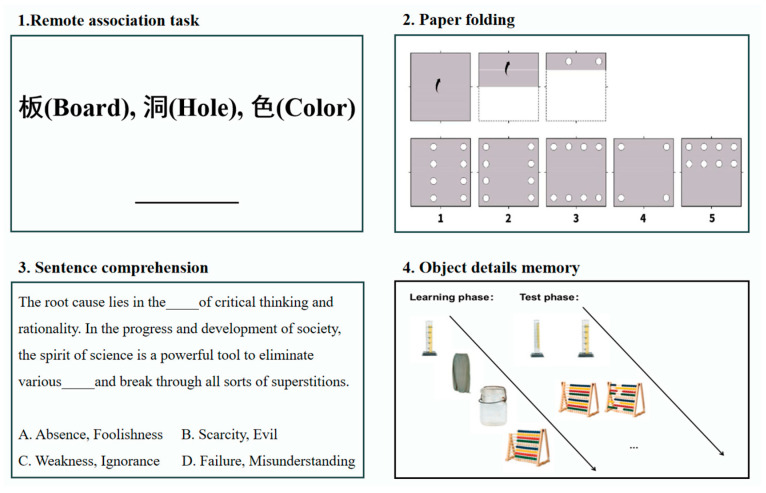
The diagrams of cognitive abilities.

**Figure 2 behavsci-15-00299-f002:**
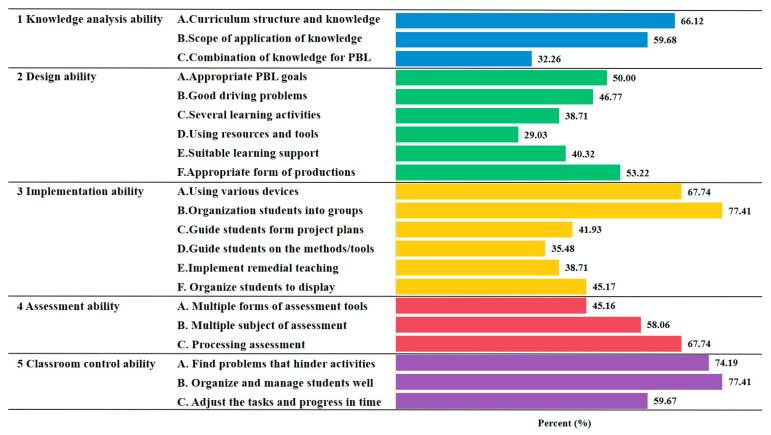
The percentage of each option in the project-based teaching questionnaire.

**Table 1 behavsci-15-00299-t001:** Describe statistics for cognitive and project-based teaching.

Task	Mean ± SD	Skewness	Kurtosis	Reliability
1. Remote association task	3.41 ± 2.84	2.21	8.14	0.77
2. Paper folding	6.17 ± 2.60	0.64	0.82	0.89
3. Sentence comprehension	5.84 ± 1.89	−1.70	−0.022	0.87
4. Object detail memory	16.32 ± 2.94	−0.88	0.34	0.96
5. Project-based teaching	0 ± 4.05	0.54	−0.47	0.84
5.1 Knowledge analysis ability	1.58 ± 0.76	0.88	−0.69	
5.2 Design ability	2.58 ± 1.54	1.05	0.17	
5.3 Implementation ability	3.06 ± 1.61	0.48	−0.74	
5.4 Assessment ability	1.71 ± 0.76	0.53	−1.04	
5.5 Classroom control ability	2.11 ± 0.77	−0.20	−1.28	

**Table 2 behavsci-15-00299-t002:** Correlation of cognitive and project-based teaching.

Task	1	2	3	4	5	5.1	5.2	5.3	5.4
1. Remote association task	-								
2. Paper folding	0.17	-							
3. Sentence comprehension	−0.14	0.25							
4. Object detail memory	0.34 **	0.23	−0.14						
5. Project-based teaching	0.03	0.34 **	0.12	−0.06					
5.1 Knowledge analysis ability	0.01	0.26 *	0.02	0.03	0.53 ***				
5.2 Design ability	0.10	0.34 **	−0.06	0.04	0.61 ***	0.69 ***			
5.3 Implementation ability	0.04	0.32 *	0.08	0.02	0.56 ***	0.54 ***	0.70 ***		
5.4 Assessment ability	0.23	0.30 *	0.09	0.14	0.36 **	0.55 ***	0.63 ***	0.51 ***	
5.5 Classroom control ability	0.10	0.38 **	0.06	0.04	0.76 ***	0.82 ***	0.90 ***	0.82 ***	0.75 ***

Note: * *p* < 0.05, ** *p* < 0.01, *** *p* < 0.001.

**Table 3 behavsci-15-00299-t003:** Regression analysis of cognitive ability’s role in project-based teaching after controlling for age, gender, teaching experience, and project-based teaching experience.

	5. Project-Based Teaching		5.1 Knowledge Analysis Ability		5.2 Design Ability		5.3 Implementation Ability		5.4 Assessment Ability		5.5 Classroom Control Ability	
	*β*	*t*	*β*	*t*	*β*	*t*	*β*	*t*	*β*	*t*	*β*	*t*
CV: Age	−0.10	−0.54	0.15	0.81	−0.09	−0.45	−0.22	−1.14	−0.15	−0.75	−0.12	−0.61
CV: Gender	0.09	0.61	0.07	0.51	0.22	1.53	−0.05	−0.35	0.05	0.36	0.06	0.42
CV: Teaching experience	0.06	0.30	0.12	0.64	0.06	0.31	0.03	0.19	0.06	0.30	−0.04	−0.22
CV: PBT experience	0.26	1.87	0.17	1.24	0.42 *	3.07	0.21	1.48	0.14	0.93	0.12	0.82
1. Remote association task	0.05	0.38	0.07	0.48	−0.01	−0.08	−0.01	−0.05	−0.01	−0.07	0.18	1.24
2. Paper folding	0.39 *	2.89	0.39 *	2.87	0.25	1.89	0.40 *	2.94	0.32	2.27	0.22	1.60
3. Sentence comprehension	−0.04	−0.25	0.13	0.91	−0.01	−0.09	−0.24	−1.65	−0.04	−0.28	0.02	0.12
4. Object detail memory	−0.09	−0.66	−0.19	−1.45	−0.06	−0.49	−0.07	−0.56	−0.06	−0.44	0.04	0.26
*R* ^2^	0.14		0.18		0.6		0.15		0.09		0.10	

Note: * *p* < 0.05, and the corrected *p* is 0.008 with the Bonferroni correction method (0.05/6 models).

**Table 4 behavsci-15-00299-t004:** Participant profiles of Study 2.

		Teacher A	Teacher B
Age (years old)		33	30
Gender		Female	Male
Teaching experience (years)		11	9
Teaching subject		Mathematics	Mathematics
Current teaching grade		Fifth grade	Fifth grade
Highest degree		Bachelor	Bachelor
Project-based teaching experience (years)		2	2
Cognitive abilities	Remote association task	2	4
	Paper folding	10	3
	Sentence comprehension	7	7
	Object detail memory	17	15

## Data Availability

Data is contained within the article.

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
