# Peer review of "The Role of Cognitive Abilities in Project-Based Teaching: A Mixed Methods Study"

_behavsci, 2025, doi:10.3390/bs15030299_

Round 1

Reviewer 1 Report

Comments and Suggestions for Authors

The article explores the relationship between teachers’ cognitive abilities and project-based teaching. It includes two studies: one using quantitative methods and the other using qualitative methods. The article focuses on an interesting topic, and the findings are important for future research. However, the current form of the article requires significant changes.

1. The article must be proofread, as there are some mistakes in language presentation, incomplete sentences, and word choice. For example, the abstract states that "this study aimed to explore the relationship between teachers' cognitive abilities and project-based teaching." It should be "teachers' cognitive abilities," not just "teachers' cognitive." The whole article should be proofread for similar mistakes.

2. The structure of the article is fine. However, it would be better if the information from both studies were presented in one consolidated section. The methods section of both studies should be combined, just as the results and discussions should be. This will help readers access the information more easily and effectively. Additionally, the cognitive abilities and the theoretical model should be addressed separately. The study selected Cognitive abilities as a variable, while the theoretical framework was designed to support the chosen hypothesis with theory.

3. The introduction and literature review lack depth in reviewing the literature on the chosen topics. Project-based learning and teaching have been widely researched in recent years. Thus, more information on the findings from the available literature would provide greater insight for readers. Furthermore, the study is situated in China. Therefore, more information on project-based teaching in China would add value to the article. What was the context of the study, such as a school or a region, along with some basic comparisons of the selected region against others, like the rationale for choosing the current location?

4. The methods should be presented together for both studies. This would provide a clear overview of the methods applied in the research. A table detailing the participants' profiles would make it easier for readers to digest the information. Additionally, more information regarding the participants in the qualitative study would be beneficial for readers. Furthermore, it should be clarified whether these two teachers were part of Study 1, and why this research chose only these two teachers.

5. The results provide good information on the study's findings. The results of both studies can be grouped under one header: Results --> Study 1 results, Study 2 results.

6. Change the title of the general discussion to just "discussion." What constitutes a general discussion? How is it different from other types of discussions? It's better to make it a discussion section.

7. Some information is missing from the article, such as the implications of the research, the conclusion, limitations, and future suggestions. These sections are crucial for the research article.

8. Delete unnecessary information, such as the heading "patent."

9. All information in the article that is not in English should be translated into English. Thus, any content provided in Chinese must include an English translation so that readers can understand the information. 

Author Response

Thank you very much for taking the time to review this manuscript. Please find the detailed responses below and the corresponding revisions in the re-submitted files.

Comment 1:

  1. The article must be proofread, as there are some mistakes in language presentation, incomplete sentences, and word choice. For example, the abstract states that "this study aimed to explore the relationship between teachers' cognitive abilities and project-based teaching." It should be "teachers' cognitive abilities," not just "teachers' cognitive." The whole article should be proofread for similar mistakes.

Response 1: We thank the reviewer for their comment. We have carefully proofread the manuscript and enlisted a professional editing service to revise it.

Comment 2:

  1. The structure of the article is fine. However, it would be better if the information from both studies were presented in one consolidated section. The methods section of both studies should be combined, just as the results and discussions should be. This will help readers access the information more easily and effectively. Additionally, the cognitive abilities and the theoretical model should be addressed separately. The study selected Cognitive abilities as a variable, while the theoretical framework was designed to support the chosen hypothesis with theory.

Response 2: Thank you for pointing this out. We agree with this comment.

(1) Therefore, we have combined the methods of Studies 1 and 2 in the sections “2. Methods: 2.1 Study 1 methods; 2.2 Study 2 methods”, and we have combined results of Studies 1 and 2 as follows:“3. Results: 3.1 Study 1 results; 3.2 Study 2 results”.

(2) We have added a separate section, “1.3 Current study”, which addresses the theoretical framework [Line 169-186].

That is, “This study aimed to explore how teachers’ cognitive abilities are associated with project-based learning and to quantitatively identify cognitive correlates of project-based teaching. The first research question is as follows: which cognitive abilities predict project-based teaching? Based on an object–spatial–verbal cognitive model (Blazhenkova & Kozhevnikov, 2009) and the importance of creativity (e.g., Ersoy, 2014), we designed Study 1 to measure cognitive abilities from four aspects: object ability, spatial ability, verbal ability, and creative ability. The relationship between these cognitive abilities and the scores of the project-based teaching questionnaire was explored in Study 1. Because these cognitive abilities are relatively independent in the cognitive model (Blazhenkova & Kozhevnikov, 2009), we hypothesized that only one or two cognitive abilities may play a unique role in project-based teaching.

To quantitatively identify the cognitive correlates of project-based teaching, we posed the second research question: how does a key cognitive predictor specifically shape the design and implementation of project-based teaching? Furthermore, by contrasting high- vs. low cognitive ability teachers’ project-based teaching plans, Study 2 could illuminate actionable strategies that align with the cognitive abilities, offering concrete recommendations for teacher education.”

Comments 3:

  1. The introduction and literature review lack depth in reviewing the literature on the chosen topics. Project-based learning and teaching have been widely researched in recent years. Thus, more information on the findings from the available literature would provide greater insight for readers. Furthermore, the study is situated in China. Therefore, more information on project-based teaching in China would add value to the article. What was the context of the study, such as a school or a region, along with some basic comparisons of the selected region against others, like the rationale for choosing the current location?

Response 3: Thank you for pointing this out. We agree with this comment.

(1) To add depth to the introduction and literature review, we made the following changes: First, we summarized the literature on project-based teaching and summarized project-based teaching in China. Second, we more clearly described the need for this study to focus on cognition. Third, we added a separate section about the theoretical framework.

(2) We added one paragraph to the Introduction to summarize project-based teaching in China and provide a related discussion in the Discussion section [Line 108-122].

That is, “This study is situated in China, where project-based learning and teaching is in a stage of rapid development. Before 2020, there was not much research on project-based teaching in China, and the total number of Chinese core journal papers published was about 50. Nearly 140 Chinese core journal papers have been published between 2020 and 2025 with the titles of "project-based teaching". This study probes into how project-based teaching works in different specific contexts, such as “Construction of interdisciplinary project-based teaching model in Geography” (Shi, Huang, & Ma, 2024), “Project-based teaching of VB program from the perspective of problem solving”(Hu, 2019) and so on. Benefiting from the new national policy of “Compulsory Education Curriculum Plan and Curriculum Standards (2022)”, project-based learning and teaching has become a new national way of compulsory education. With many teachers needing to carry out project-based teaching, ways to better cultivate their project-based teaching are urgent. Despite extensive studies conducted in China and other countries, there remains a gap in the literature regarding the role of teachers’ cognitive abilities in project-based teaching.”

“Project-based teaching has not been popular in China for long, so most teachers lack extensive experience with project-based learning. It is precisely because Chinese teachers need more relevant training that the current study is significant for project-based teaching training in China.”

(3) We have added more information about the context of this study [212-222]. That is, “This study included 62 primary school teachers from more than 40 schools in Beijing and Hebei, China. These regions started project-based teaching earlier than most regions in China and are more representative. ”

Comments 4:

  1. The methods should be presented together for both studies. This would provide a clear overview of the methods applied in the research. A table detailing the participants' profiles would make it easier for readers to digest the information. Additionally, more information regarding the participants in the qualitative study would be beneficial for readers. Furthermore, it should be clarified whether these two teachers were part of Study 1, and why this research chose only these two teachers.

Response 4: Thank you for pointing this out. We agree with this comment.

(1) We have combined methods of Studies 1 and 2 as follows: “2. Methods: 2.1 Study 1 methods; 2.2 Study 2 methods”.

(2) We provided the profiles of the participants in Study 2 in Table 1. We have included information about the participants’ highest degree and experience with project-based teaching in Study 1 and the current teaching grade, highest degree, and experience with project-based teaching of participants in in Study 2.

(3) We clarified that the two teachers involved in Study 2 were part of Study 1, and we have provided a justification for choosing these two teachers [Line 297-305].

That is, “After the data were collected in Study 1, the teachers began to develop project-based teaching plans, but only less than 10 teachers completed their final plans. Therefore, Study 2 selected two teachers from Study 1 participants with different spatial abilities and completed the final plans. Teacher A had a good spatial ability (paper folding score of 10), and teacher B had a lower spatial ability (paper folding score of 3), but there were no differences in the other aspects; see details in Table 1. For example, they were both mathematics teachers from Beijing, China. There were no differences in the other cognitive abilities, including remote association tasks, sentence comprehension, and object detail memory.”

Comments 5:

  1. The results provide good information on the study's findings. The results of both studies can be grouped under one header: Results --> Study 1 results, Study 2 results.

Response 5: Thank you for pointing this out. We have combined results of Studies 1 and 2 as follows: “3. Results: 3.1 Study 1 results; 3.2 Study 2 results”.

Comments 6:

  1. Change the title of the general discussion to just "discussion." What constitutes a general discussion? How is it different from other types of discussions? It's better to make it a discussion section.

Response 6: Thank you for pointing this out. We have changed the title “General discussion” to just “Discussion”.

Comments 7:

  1. Some information is missing from the article, such as the implications of the research, the conclusion, limitations, and future suggestions. These sections are crucial for the research article.

Response 7: Thank you for pointing this out. We have added subheadings to the previous Discussion. These are “4.3 Implications” and “4.4 Limitations and future suggestions”. We have also provided a new section: “5. Conclusions” [Line 604-612].

Comments 8:

  1. Delete unnecessary information, such as the heading "patent."

Response 8: Thank you for pointing this out. We have deleted “Patent”.

Comment 9:

  1. All information in the article that is not in English should be translated into English. Thus, any content provided in Chinese must include an English translation so that readers can understand the information.

Response 9: Thank you for pointing this out. We have translated Figure 1 and related examples in the manuscript into English.

Reviewer 2 Report

Comments and Suggestions for Authors

This is an interesting paper about the cognitive abilities of teachers who engage in project-based learning. Specifically, the paper conceptualises what such teachers do as ‘project-based teaching’ (PBT) and uses the ‘object-spatial-verbal’ cognitive model (OSV) to interrogate teachers’ cognitive abilities. The first study concludes that it is teachers’ spatial ability that plays an important role in PBT, which is further explored in qualitative terms in the second study. The findings of the paper are used to argue that teachers should be encouraged to use schemata for PBT, and that assessment systems might be developed for teachers’ cognitive (and specifically spatial) abilities.

In my view, much of the paper seems sound. The underlying study is interesting, and the paper is fairly well-written. Some aspects would benefit from more explanation, however. I therefore suggest that the authors be requested to address the following points:

·         The Introduction would benefit from more explicit argumentation about why this study focusses on cognitive ability. The text does argue that teachers are not sufficiently studied in the literature on problem-based learning. But this does not necessarily mean that we should study teachers using the lens of *cognition*. Why is cognition the angle here? And why was the OSV framework chosen to interrogate this cognition?

·         The paper spends a lot of time arguing that PBT has not been much studied before. But then a range of papers that do study PBT are cited on lines 71-76. In my view, this content should be expanded into a more full paragraph. Specifically, it should be argued why a paper focussed on cognition can add something new to this literature.

·         It would be useful also in the Introduction to briefly explain how the OSV framework influences the two studies that are reported.

·         For both studies, it would be useful at the beginning (lines 106-122 and 259-265) to explain in more detail the aim and rationale of the study. Simply stating that something has not been studied before is insufficient. What is the positive motivation for the study, more specifically?

·         In sections 2.1.1 and 3.1.1 we need to read more about the level of experience of PBT that the participants have. Stating how long they have been teaching in general is interesting. But surely it is their experience *with PBT specifically* which is important here?

·         In section 2.1.2, it would be useful to explain the relationships between the tasks and the OSV model. At present, the text says that these tasks are aligned, but does not explain how.

·         On page 7, the authors move to analysing issues of age and gender. This has not been previously highlighted as a core focus of the study. This should either be justified in more depth as a focus in the earlier stages of the paper, or else this part of the analysis could be deleted.

·         In section 3.1.2, it would be useful to explain how the use of the template exercise was influenced by the OSV framework.

·         Lines 371-400 do a good job at explaining the implications for practice of this study. But what we need to read more of is the contributions to research. In what ways do the findings in this paper present new insights which would be interesting to the authors of the papers reviewed in section 1? I think we need an extra paragraph about this issue.

Author Response

Thank you very much for taking the time to review this manuscript. Please find the detailed responses below and the corresponding revisions in the re-submitted files.

Comment 1:

The Introduction would benefit from more explicit argumentation about why this study focusses on cognitive ability. The text does argue that teachers are not sufficiently studied in the literature on problem-based learning. But this does not necessarily mean that we should study teachers using the lens of *cognition*. Why is cognition the angle here? And why was the OSV framework chosen to interrogate this cognition?

Response 1: Thank you for pointing this out. We agree with this comment.

(1) We have added a paragraph to explain why we need to take a cognitive perspective [Line 124-136].

That is, “Cognitive ability refers to any ability that concerns the processing of mental information (Carroll, 1993). Such cognitive abilities are foundational to complex tasks. For example, studies on problem-solving highlight that spatial and verbal abilities are critical for representing abstract relationships and coordinating multi-step processes (Hegarty & Kozhevnikov, 1999; Newcombe & Shipley, 2015; Wang et al., 2022). As a complex teaching task, project-based teaching may also require cognitive abilities. For example, teachers with higher spatial ability demonstrate superior curriculum integration in STEM fields (Uttal et al., 2013). Meanwhile, teachers with stronger verbal abilities excel in scaffolding student discussions (Hattie, 2012). Crucially, cognitive abilities are not merely “static traits” but malleable skills that can be enhanced through targeted training (Hawes et al., 2022; Rogge et al., 2017; Uttal et al., 2013). By examining teachers’ cognition, this study seeks to uncover actionable pathways for professional development..”

(2) We provided an explanation for why we choose the object–spatial–verbal cognitive model [Line 156-164].

That is, “The object–spatial–verbal cognitive model clearly distinguishes the function of three types of cognitive systems from the basis of brain regions (Blazhenkova & Kozhevnikov, 2009). For example, spatial ability is associated with the dorsal stream (Cona & Scarpazza, 2019), verbal ability is dependent on the left temporal lobe (Broca, 1861; Heyer et al., 2022), and object processing is dependent on the ventral stream (Ayzenberg & Behrmann, 2022; Cabeza & Nyberg, 2000). It justified testing each ability’s unique contribution to project-based teaching rather than aggregating them into a single “general intelligence” metric. This helps to accurately identify key cognitive components in project-based teaching.”

Comments 2:

The paper spends a lot of time arguing that PBT has not been much studied before. But then a range of papers that do study PBT are cited on lines 71-76. In my view, this content should be expanded into a more full paragraph. Specifically, it should be argued why a paper focused on cognition can add something new to this literature.

Response 2: (1) We apologize for the lack of clarity. Our original intention was to say that there is less research on project-based teaching than there is on project-based learning. To better convey this, we have reorganized the relevant sentences in Introduction [Line 83-85]. That is, “Compared with the considerable body of literature on project-based learning (Hung et al., 2019; Kokotsaki et al., 2016; Zhang & Ma, 2023), relatively little attention has been paid to project-based teaching.” 

(2) We have expanded 71-76 into a more full paragraph in the Introduction [Line 92-107].

That is, “Previous studies on actual project-based teaching explored a wide range of influencing factors and aspects related to its implementation and effectiveness. For instance, research has examined how gender, implementation experience, and school management impact project-based teaching (Gomez-Pablos et al., 2017). Attitudes towards project-based learning have also been a focus, with findings indicating that positive attitudes can significantly enhance student engagement and learning outcomes (Habok & Nagy, 2016). Additionally, multimedia resources are a crucial factor in supporting project-based teaching, as they can provide students with more interactive and immersive learning experiences (Ozdamli, 2011). Teacher motivation has been identified as another key factor, with motivated teachers being more likely to effectively implement project-based teaching and support student learning (Lam et al., 2009). Professional identity has been highlighted as an important aspect, with teachers with a strong professional identity being better equipped to navigate the challenges of project-based teaching (Tsybulsky & Muchnik-Rozanov, 2019).”

  • In the Discussion, we have included an explanation of how the cognitive perspective provides something new to the literature[Line 514-527].

That is, “In the field of project-based teaching, previous studies on teachers’ project-based learning have explored the influencing factors of gender, implementation experience, and school management (Gomez-Pablos et al., 2017); professional experience and teaching direction (Rogers et al., 2011); attitudes towards project-based learning (Habok & Nagy, 2016); multimedia resources (Ozdamli, 2011); teacher motivation (Lam et al., 2009); professional identity (Tsybulsky & Muchnik-Rozanov, 2019); and more. Unlike these studies, our study aimed to fill the gap in the literature regarding the role of teachers’ cognitive abilities in project-based teaching. It uncovered an actionable pathway for professional development from a cognitive perspective. In terms of cognitive studies, this study provides new evidence for the practical application of the object–spatial–verbal cognitive model. The results of spatial ability alone have a significant predictive effect, indirectly demonstrating the separation and conflict between object, spatial, and verbal abilities. Moreover, the current results demonstrate the particularity and importance of spatial ability in many cognitive abilities.”

Comments 3:

It would be useful also in the Introduction to briefly explain how the OSV framework influences the two studies that are reported.

Response 3: Thank you for pointing this out. We agree with this comment. Therefore, we have provided an explanation of how the OSV framework influences the two studies [Line 169-186].

That is “This study aimed to explore how teachers’ cognitive abilities are associated with project-based learning and to quantitatively identify cognitive correlates of project-based teaching. The first research question is as follows: which cognitive abilities predict project-based teaching? Based on an object–spatial–verbal cognitive model (Blazhenkova & Kozhevnikov, 2009) and the importance of creativity (e.g., Ersoy, 2014), we designed Study 1 to measure cognitive abilities from four aspects: object ability, spatial ability, verbal ability, and creative ability. The relationship between these cognitive abilities and the scores of the project-based teaching questionnaire was explored in Study 1. Because these cognitive abilities are relatively independent in the cognitive model (Blazhenkova & Kozhevnikov, 2009), we hypothesized that only one or two cognitive abilities may play a unique role in project-based teaching.

To quantitatively identify the cognitive correlates of project-based teaching, we posed the second research question: how does a key cognitive predictor specifically shape the design and implementation of project-based teaching? Furthermore, by contrasting high- vs. low cognitive ability teachers’ project-based teaching plans, Study 2 could illuminate actionable strategies that align with the cognitive abilities, offering concrete recommendations for teacher education.”

Comments 4:

For both studies, it would be useful at the beginning (lines 106-122 and 259-265) to explain in more detail the aim and rationale of the study. Simply stating that something has not been studied before is insufficient. What is the positive motivation for the study, more specifically?

Response 4: Thank you for pointing this out. First, we have provided the research questions, objectives, and hypotheses of the two studies in the Introduction [Line 169-186]. Please see the details in Comment 3. We have also explained the method of the two studies in greater detail in the Methods section [Line 193-210; Line 287-295].

That is, “Study 1 aimed to explore the relationship between cognitive abilities and project-based teaching. Four cognitive abilities were measured—object ability, spatial ability, verbal ability, and creative ability—and a project-based teaching questionnaire was conducted in this study. Object ability was measured using an object detail memory task (Brady et al., 2008), spatial ability was measured using a paper folding task (Ekstrom et al., 1976), and verbal ability was measured using a sentence comprehension task (Cui et al., 2023). The creative ability was measured using a remote association task (Mednick, 1962). The project-based teaching questionnaire was adopted from a related master’s thesis in China (Lu, 2019). Moreover, we controlled for important covariates to explore the unique role of cognition in this study, such as age, gender, teaching experience, and project-based teaching experience. For example, meta-analysis evidence suggests that males tend to outperform females in spatial tasks (Lauer, Yhang & Lourenco, 2019; Linn & Petersen, 1985). Cognitive abilities (e.g., spatial ability and working memory) typically decline with age, particularly after early adulthood (Salthouse, 2010). While teaching experience and project-based teaching experience are directly associated with project-based teaching, we expect that only some cognitive abilities play a unique role in project-based teaching after controlling for age, gender, teaching experience, and project-based teaching experience.”

“In Study 1, we found that among all the cognitive abilities, only spatial ability was associated with project-based teaching. However, relying on self-reported measures for project-based teaching may introduce bias, as actual competencies might differ from perceived abilities. Moreover, quantitative correlations (Study 1) cannot reveal why spatial ability matters. Therefore, in Study 2, we chose project-based teaching plans completed by two teachers with different spatial abilities to further demonstrate the relationship between spatial ability and project-based teaching. This study can supplement the deficiency of quantitative research in Study 1 through qualitative research.”

Comments 5:

In sections 2.1.1 and 3.1.1 we need to read more about the level of experience of PBT that the participants have. Stating how long they have been teaching in general is interesting. But surely it is their experience *with PBT specifically* which is important here?

Response 5: Thank you for pointing out an interesting variable. The teachers in Study 1 were exposed to project-based teaching for approximately 2 years, and the two teachers in Study 2 were also exposed to project-based teaching for approximately 2 years. 

(1) We provided relevant information about the participants in the Methods section [Line 216].

(2) We controlled for project-based teaching experience as an important covariate in the regression analysis of Study 1 [Line 389-403]. Results showed that project-based teaching experience significantly predicted design ability, with β = .42 and p =.003. This shows that the design of project-based learning is closely related to prior experience.

(3) We have included a discussion about the experience of project-based teaching in the Discussion section [Line 485-493].

That is, “On the other hand, the 2 years of experience with project-based teaching is significant as it suggests that the teachers had sufficient time to develop and refine their practices within a project-based teaching framework. However, it also implies that these teachers were still adapting to and integrating project-based teaching into their teaching methods, which may influence the findings.”

Comments 6:

In section 2.1.2, it would be useful to explain the relationships between the tasks and the OSV model. At present, the text says that these tasks are aligned, but does not explain how.

Response 6: Thank you for pointing this out. We have provided more detailed information about each task in the Methods section.

That is, [Line 254-256] Paper folding: This task estimated spatial ability, especially spatial visualization (Ekstrom et al., 1976). This corresponds to the spatial dimension of the object–spatial–verbal model, which involves the ability to understand and navigate spatial configurations.

[Line 261-263] Sentence comprehension: This test measured verbal ability (Cui et al., 2023). This aligns with the verbal dimension of the object–spatial–verbal model, which focuses on the comprehension and production of linguistic information.

[Line 271-274] Object detail memory: The test measured the ability to remember details of objects (Brady et al., 2008). This aligns with the object processing dimension of the object–spatial–verbal model, which focuses on the detailed and accurate representation of visual stimuli.”

Comments 7:

On page 7, the authors move to analysing issues of age and gender. This has not been previously highlighted as a core focus of the study. This should either be justified in more depth as a focus in the earlier stages of the paper, or else this part of the analysis could be deleted.

Response 7: Thank you for pointing this out.

(1) We have added reasons for using the analysis age and gender as covariates [Line 203-208]. That is “…meta-analysis evidence suggests that males tend to outperform females in spatial tasks (Lauer, Yhang & Lourenco, 2019; Linn & Petersen, 1985). Cognitive abilities (e.g., spatial ability and working memory) typically decline with age, particularly after early adulthood (Salthouse, 2010).”

(2) We have included more covariates (i.e., teaching experience and project-based teaching experience) in the regression analysis to increase its necessity [Line 207-210; Line389-403]. Teaching experience and project-based teaching experience are directly associated with project-based teaching. The results of regression did not change significantly after adding these control variables; only providing new results regarding the project-based teaching experience significantly predicted the design ability.

Comments 8:

In section 3.1.2, it would be useful to explain how the use of the template exercise was influenced by the OSV framework.

Response 8: Thank you for pointing this out. The template in Study 2 was designed according to the five core elements of project-based teaching, rather than the object–spatial–verbal cognitive framework. Therefore, we have connected the template and five core elements of project-based teaching [Line 320-324].

That is, “This template was designed according to the five core elements of project-based teaching given in this study. As the plans were presented in text form, the classroom control ability was not analyzed in this study. Therefore, this template includes four parts: textbook knowledge analysis, project-based learning design, implementation procedures, and assessment plans (see details in Appendix 2).”

Comments 9:

Lines 371-400 do a good job at explaining the implications for practice of this study. But what we need to read more of is the contributions to research. In what ways do the findings in this paper present new insights which would be interesting to the authors of the papers reviewed in section 1? I think we need an extra paragraph about this issue.

Response 9: Thank you for pointing this out. We agree with this comment. Therefore, we have added one paragraph in Discussion to summarize the contribution of current study [Line 514-527].

That is, “In the field of project-based teaching, previous studies on teachers’ project-based learning have explored the influencing factors of gender, implementation experience, and school management (Gomez-Pablos et al., 2017); professional experience and teaching direction (Rogers et al., 2011); attitudes towards project-based learning (Habok & Nagy, 2016); multimedia resources (Ozdamli, 2011); teacher motivation (Lam et al., 2009); professional identity (Tsybulsky & Muchnik-Rozanov, 2019); and more. Unlike these studies, our study aimed to fill the gap in the literature regarding the role of teachers’ cognitive abilities in project-based teaching. It uncovered an actionable pathway for professional development from a cognitive perspective. In terms of cognitive studies, this study provides new evidence for the practical application of the object–spatial–verbal cognitive model. The results of spatial ability alone have a significant predictive effect, indirectly demonstrating the separation and conflict between object, spatial, and verbal abilities. Moreover, the current results demonstrate the particularity and importance of spatial ability in many cognitive abilities.”

Reviewer 3 Report

Comments and Suggestions for Authors

Few concerns. Study seems well set-up, structured, and reported. Minor points--good paper overall.

Literature review could use some updating, especially on spatial ability.

Regression analysis was exploratory and included 6 different models. The chance of errors increase when repeating a statistical test on the same variables. 

Section 3.1.1. Can you explain the cutoff scores? Binning data is an important part of research. An exceptionally small sample, even for a case study. A significant limitation is representativeness; 2 people probably cannot speak for the entire group. 

3.1.2 does not actually discuss any analysis; how was the data stored, analyzed, and compared?

Author Response

Thank you very much for taking the time to review this manuscript. Please find the detailed responses below and the corresponding revisions in the re-submitted files.

Comments 1:

Literature review could use some updating, especially on spatial ability.

Response 1: Thank you for pointing this out. We agree with this comment. We have referenced an additional 10 studies published in the last 5 years.

That is, “Finland’s new National Curriculum for 2016 emphasizes interdisciplinary and integrated education to dismantle subject barriers and prepare students for future needs through project-based learning (Braskén, Hemmi & Kurtén, 2020)”; “This study probes into how project-based teaching works in different specific contexts, such as “Construction of interdisciplinary project-based teaching model in Geography” (Shi, Huang, & Ma, 2024); “Project-based teaching of VB program from the perspective of problem solving”(Hu, 2019) and so on”; “…meta-analysis evidence suggests that males tend to outperform females in spatial tasks (Lauer, Yhang & Lourenco, 2019; Linn & Petersen, 1985)”; “Compared with the considerable body of literature on project-based learning (Hung et al., 2019; Kokotsaki et al., 2016; Zhang & Ma, 2023), there relatively little attention has been paid to project-based teaching”; “Spatial ability reflects an understanding of more general, abstract form of pictorial expressions (Hegarty, 2010; Tiwari, Shah & Muthiah, 2024)”; “Crucially, cognitive abilities are not merely “static traits” but malleable skills that can be enhanced through targeted training (Hawes, et al., 2022; Rogge et al., 2017; Uttal et al., 2013).”; “For example, spatial ability is associated with the dorsal stream (Cona & Scarpazza, 2019), verbal ability is dependent on the left temporal lobe (Broca, 1861; Heyer et al., 2022), and object processing is dependent on the ventral stream (Ayzenberg & Behrmann, 2022; Cabeza & Nyberg, 2000).” 

Comments 2:

Regression analysis was exploratory and included 6 different models. The chance of errors increase when repeating a statistical test on the same variables.

Response 2: Thank you for pointing this out. We agree with this comment. Therefore, we have included the Bonferroni correction in the regression to control the errors [Line 389-403]. Statistical significance was set at a Bonferroni corrected p-value of .008 (0.05 / 6 because there were six models). The result shows that the assessment ability is no longer significant, which we have corrected in the Results section.

Comments 3:

Section 3.1.1. Can you explain the cutoff scores? Binning data is an important part of research. An exceptionally small sample, even for a case study. A significant limitation is representativeness; 2 people probably cannot speak for the entire group.

Response 3: Thank you for pointing this out.

(1) In fact, the two teachers in Study 2 were sampled for convenience. After the data collection in Study 1, all teachers began to develop project-based teaching plans, but fewer than 10 teachers completed the final plans. Therefore, for Study 2, we selected two teachers from among the Study 1 participants with different spatial abilities and completed the final plans for further analysis. Teacher A had good spatial ability (paper folding: 10) and teacher B had less spatial ability (paper folding: 3), but there were no other differences (see the details in Table 1). For example, they were both math teachers from Beijing, China. There were no differences in their other cognitive abilities, including remote association, sentence comprehension, and object detail memory. 

However, with a limited sample, we did not find other teachers with differences in spatial ability who did not also have other differences. In future studies, the sample size needs to be further expanded to find matching cases.

(2) We have outlined the limitation of representativeness in the Discussion [Line 509-586].

That is, “…the sample size was small. With only 62 primary school teachers involved, the findings may not be broadly generalizable to the larger population of educators. The two teachers in Study 2 probably cannot speak for the entire group, as this sample size does not account for variations in teaching styles, educational contexts, or individual teacher characteristics. A smaller sample size can limit the statistical power of the study, potentially leading to less reliable and less robust results. Future research should aim to include a more extensive and diverse sample of teachers to enhance the generalizability and reliability of the findings. 

Comments 4:

3.1.2 does not actually discuss any analysis; how was the data stored, analyzed, and compared?

Response 4: Thank you for pointing this out. We have added the information of data stored, analyzed, and compared [Line 325-330].

That is “These documents were stored in anonymized digital formats.” “This study mainly explores the embodiment of spatial ability in project-based teaching plans. Therefore, the analysis process mainly extracts information related to spatial ability, such as the use of schemata (Boonen et al., 2013; Boonen et al., 2014; Hegarty & Kozhevnikov, 1999), abstract concepts (Hegarty, 2010), overall plan (Gonçalves & Ferreira, 2015), and more. Differences were found through a qualitative comparison of the two plans.”

Reviewer 4 Report

Comments and Suggestions for Authors

Dear authors:

I think that the issue under investigation is relevant. However, your manuscript has shortcomings. It seems to me that some of them have to do with the structure of the manuscript.

Here are my suggestions/comments:

In the abstract: 

-it is clear that 'this study aimed to explore the relationship between teachers' cognitive and project-based teaching'.

-We don't know what kind of study was done (what was the research method/methodology?). 

So, if we triangulate the abstract with the title and sections of the manuscript, it doesn't seem to be a complete relationship.

I suggest

- Create a subsection after the "Introduction" with the research problem, objective(s), hypotheses (?).

- Include a section on the research methodology. 

For example:

We need to understand what kind of research has been done. 

Find out if the questionnaire has been validated. Is it a questionnaire or a survey? You write questionnaire and then, for example in section 5, you write survey.

What criteria were used to select participants for Study 1 and Study 2?

"3.1.2 Data collection and analysis" does not provide any references to give scientific soundness to what they describe and have done. It should be noted that in the "References" section I found only 4 references from the last 4 years. This should be improved.

-I suggest that Figure 1 be written entirely in English.

I don't understand why there is a section '4. General discussion' when there is a subsection '3.2 Results and discussion'. Perhaps this structure is the reason why we can't find the conclusions of the study written clearly and precisely. In fact, it seems to me that this sentence from the abstract “This study indicated that a teacher’s spatial ability is closely associated with their project-based teaching; and it provides a new perspective for teachers to cultivate project-based teaching.” ' should be properly supported in the last section of the article, as the conclusion of the study.

Also, about the conclusions of the research, it seems to me that they make generalisations that are questionable given the limitations of this research. For example: “Additional training focused on spatial ability will be incorporated into teacher professional development programs concerning project-based teaching.”

Rew

Author Response

Thank you very much for taking the time to review this manuscript. Please find the detailed responses below and the corresponding revisions in the re-submitted files.

Comments 1:

In the abstract:

-it is clear that 'this study aimed to explore the relationship between teachers' cognitive and project-based teaching'.

-We don't know what kind of study was done (what was the research method/methodology?).

So, if we triangulate the abstract with the title and sections of the manuscript, it doesn't seem to be a complete relationship.

I suggest

- Create a subsection after the "Introduction" with the research problem, objective(s), hypotheses (?).

- Include a section on the research methodology.

For example:

We need to understand what kind of research has been done.

Response 1: Thank you for pointing this out. We agree with this comment.

(1) We have modified the Abstract accordingly [Line 12-13; Line 18-21]. That is, “This study aimed to explore the relationship between teachers’ cognitive abilities and project-based teaching using a mixed-methods design. In Study 1, a quantitative correlational analysis was conducted with 62 primary school teachers.” “ In Study 2, a qualitative comparative case study was employed to examine how spatial ability manifests in two teachers’ project-based teaching plans. ”

(2) We have added a section, “1.3 Current study”, to address the research objectives, questions and hypotheses [Line 169-186].

That is, “This study aimed to explore how teachers’ cognitive abilities are associated with project-based learning and to quantitatively identify cognitive correlates of project-based teaching. The first research question is as follows: which cognitive abilities predict project-based teaching? Based on an object–spatial–verbal cognitive model (Blazhenkova & Kozhevnikov, 2009) and the importance of creativity (e.g., Ersoy, 2014), we designed Study 1 to measure cognitive abilities from four aspects: object ability, spatial ability, verbal ability, and creative ability. The relationship between these cognitive abilities and the scores of the project-based teaching questionnaire was explored in Study 1. Because these cognitive abilities are relatively independent in the cognitive model (Blazhenkova & Kozhevnikov, 2009), we hypothesized that only one or two cognitive abilities may play a unique role in project-based teaching.

To quantitatively identify the cognitive correlates of project-based teaching, we posed the second research question: how does a key cognitive predictor specifically shape the design and implementation of project-based teaching? Furthermore, by contrasting high- vs. low cognitive ability teachers’ project-based teaching plans, Study 2 could illuminate actionable strategies that align with the cognitive abilities, offering concrete recommendations for teacher education. ” 

(3) We have included the research methodology in the first paragraph of the Methods section [Line 188-191].

That is, “The current study emphasized measurable cognitive constructs (Study 1) while acknowledging specific role of cognitive ability in project-based teaching through qualitative exploration (Study 2). Quantitative data identified key cognitive predictors, and qualitative data elucidated how these abilities operate in practice. ”

Comment 2:

Find out if the questionnaire has been validated. Is it a questionnaire or a survey? You write questionnaire and then, for example in section 5, you write survey.

Response 2: We apologize for the lack of clarity.

(1) The questionnaire was validated in 213 Chinese primary school teachers. This study found that its reliability and validity were good, with Cronbach’s α coefficient of 0.97 and a correlation of the total score and each question greater than 0.40 (Lu, 2019).

(2) This study used a questionnaire rather than a survey. We have standardized terms throughout the manuscript, referring to a questionnaire.

Comment 3:

What criteria were used to select participants for Study 1 and Study 2?

Response 3: Study 1 included 62 primary school teachers from more than 40 schools in Beijing and Hebei, China. These teachers had voluntarily enrolled in a teacher training program.

After data collection in Study 1, all teachers began to develop project-based teaching plans, but only a few teachers completed the final plans. Therefore, Study 2 selected two teachers from among the Study 1 participants with different spatial abilities and completed the final plans for further analysis. Teacher A, who had good spatial ability (paper folding: 10) and teacher B had less spatial ability (paper folding: 3); there were no differences in other aspects (see Table 1 for details). Both were math teachers from Beijing, China. There were no differences identified in their other cognitive abilities, including remote association, sentence comprehension, and object detail memory.

We have included the above information in Method section [Line 297-305].

Comment 4:

"3.1.2 Data collection and analysis" does not provide any references to give scientific soundness to what they describe and have done.

Response 4: Thank you for pointing this out. We have added references for Study 2, data analysis [Line 325-330].

That is “This study mainly explores the embodiment of spatial ability in project-based teaching plans. Therefore, the analysis process mainly extracts information related to spatial ability, such as the use of schemata (Boonen et al., 2013; Boonen et al., 2014; Hegarty & Kozhevnikov, 1999), abstract concepts (Hegarty, 2010), overall plan (Gonçalves & Ferreira, 2015), and more. Differences were found through a qualitative comparison of the two plans.”

Comments 5:

It should be noted that in the "References" section I found only 4 references from the last 4 years. This should be improved.

Response 5: Thank you for pointing this out. We have added 10 studies published in the last 5 years.

That is, “Finland’s new National Curriculum for 2016 emphasizes interdisciplinary and integrated education to dismantle subject barriers and prepare students for future needs through project-based learning (Braskén, Hemmi & Kurtén, 2020)”; “This study probes into how project-based teaching works in different specific contexts, such as “Construction of interdisciplinary project-based teaching model in Geography” (Shi, Huang, & Ma, 2024); “Project-based teaching of VB program from the perspective of problem solving”(Hu, 2019) and so on”; “…meta-analysis evidence suggests that males tend to outperform females in spatial tasks (Lauer, Yhang & Lourenco, 2019; Linn & Petersen, 1985)”; “Compared with the considerable body of literature on project-based learning (Hung et al., 2019; Kokotsaki et al., 2016; Zhang & Ma, 2023), there relatively little attention has been paid to project-based teaching”; “Spatial ability reflects an understanding of more general, abstract form of pictorial expressions (Hegarty, 2010; Tiwari, Shah & Muthiah, 2024)”; “Crucially, cognitive abilities are not merely “static traits” but malleable skills that can be enhanced through targeted training (Hawes, et al., 2022; Rogge et al., 2017; Uttal et al., 2013).”; “For example, spatial ability is associated with the dorsal stream (Cona & Scarpazza, 2019), verbal ability is dependent on the left temporal lobe (Broca, 1861; Heyer et al., 2022), and object processing is dependent on the ventral stream (Ayzenberg & Behrmann, 2022; Cabeza & Nyberg, 2000).”” 

Comments 6:

-I suggest that Figure 1 be written entirely in English.

Response 6: Thank you for pointing this out. We have translated Figure 1 and related examples in the manuscript into English.

Figure 1. The diagrams of cognitive abilities.

Comments 7:

I don't understand why there is a section '4. General discussion' when there is a subsection '3.2 Results and discussion'. Perhaps this structure is the reason why we can't find the conclusions of the study written clearly and precisely. In fact, it seems to me that this sentence from the abstract “This study indicated that a teacher’s spatial ability is closely associated with their project-based teaching; and it provides a new perspective for teachers to cultivate project-based teaching.” ' should be properly supported in the last section of the article, as the conclusion of the study.

Response 7: Thank you for pointing this out.

(1) We have reorganized the structure of the article. We have combined the methods of Studies 1 and 2 as follows: “2.Methods: 2.1 Study 1 methods; 2.2 Study 2 methods”. We have combined the results of Studies 1 and 2 as follows: “3. Results: 3.1 Study 1 results; 3.2 Study 2 results”. We have change the “General discussion” to just “Discussion”.

(2) We have added a new section,  “5. Conclusion”, which summarizes the conclusions of current studies [Line 604-612].

That is, “In conclusion, this study underscores the critical role of teachers' spatial ability in project-based teaching. Through a comprehensive approach involving both quantitative and qualitative analyses, this study revealed that among various cognitive abilities tested, spatial ability is the most significant predictor of teachers' project-based teaching. These findings suggest that teachers with higher spatial abilities are better equipped to guide and support students in project-based learning environments. This study not only highlights the importance of spatial ability in teaching but also offers a novel perspective for teacher training and professional development.”

Comments 8:

Also, about the conclusions of the research, it seems to me that they make generalizations that are questionable given the limitations of this research. For example: “Additional training focused on spatial ability will be incorporated into teacher professional development programs concerning project-based teaching.”

Response 8: Thank you for pointing this out. We agree with this comment. Therefore, we made our inferences more carefully [Line 597-603]. That is, “…while our study suggests that incorporating additional training focused on spatial ability into teacher professional development programs could be beneficial, we recognize the limitations of our research. Future studies should aim to validate these findings with more extensive and diverse samples and explore the long-term impact of spatial ability on teaching practices. Furthermore, professional development programs should be designed to address the specific needs of teachers in different educational contexts.”

Reviewer 5 Report

Comments and Suggestions for Authors

This text tries to analyze what are the cognitive abilities needed for project based teaching. It presents a study that links teachers' perceptions about their own project-based teaching abilities to 4 types of cognitive abilities. Additionally, it contrasts the performance of two teachers while designing a project-based learning activity. All of this is based only on the idea that there is little research about teachers' characteristics in project-based teaching. A specific need for this research, derived from a state of the art analysis is not presented. There is not a clear argumentative line to follow. 

Additionally, English is very poor. 

I put here some comments about the structure, but it's really difficult to understand the text.

The first paragraph needs references for all what is said in there.

In section 1.1, Hung et al., (2019) cannot be used as a reference for project based teaching because authors talk about project based learning. 

You cannot take out some Buck Institute's standards just by saying "we think" (line 61). You have to prove this. 

It is not clear why you are looking to search for a link between cognitive abilities and project-based teaching. 

There is no discussion for any of the studies. Results should be contrasted to those obtained by other authors to know if they are good or new. 

English should be checked. There are phrases that cannot be understood. For example "More importantly, as cognitive ability is an important ability in human, there is no research to explore the role of cognitive abilities on project-based teaching", or "Project-based teaching as another major subject of project-based learning, it also needs to be paid attention to.", or "In teaching class, the bulid of culture is not formed in one day, nor is it the content that must be completed in every class of project-based teaching, so we do not think it is the core content."

Comments on the Quality of English Language

This text needs serious work. Authors don't present a strong argumentation to know that this work will extend the frontier of knowledge. There seems that there is no need for this research. 

Additionally, the writing is so poorly executed that it does not allow understanding of the text.

Author Response

Thank you very much for taking the time to review this manuscript. Please find the detailed responses below and the corresponding revisions in the re-submitted files.

Comments 1:

This text tries to analyze what are the cognitive abilities needed for project based teaching. It presents a study that links teachers' perceptions about their own project-based teaching abilities to 4 types of cognitive abilities. Additionally, it contrasts the performance of two teachers while designing a project-based learning activity. All of this is based only on the idea that there is little research about teachers' characteristics in project-based teaching. A specific need for this research, derived from a state of the art analysis is not presented. There is not a clear argumentative line to follow.

Response 1: Thank you for pointing this out. We added more information to the Introduction to make the argumentative line clearer [Line 40-48].

That is, “Previous studies explored contextual and affective factors of project-based teaching (e.g., school policies and teacher motivation; Gomez-Pablos et al., 2017; Lam et al., 2009). However, cognitive abilities are foundational to complex tasks (Hegarty & Kozhevnikov, 1999; Newcombe & Shipley, 2015; Wang et al., 2022). As a complex teaching task, project-based teaching may also require cognitive abilities. Currently, no studies have explored the role of cognitive abilities in project-based teaching. Therefore, this study aims to investigate how teachers’ cognitive abilities are associated with project-based learning.” 

Comments 2:

Additionally, English is very poor.

Response 2: Thank you for pointing this out. We have carefully proofread the manuscript, and we have enlisted a professional English editing service to revise it.

Comments 3:

I put here some comments about the structure, but it's really difficult to understand the text.

Response 3: Thank you for pointing this out. We have reorganized the structure of the article. We have combined the methods of Studies 1 and 2 as follows: “2.Methods: 2.1 Study 1 methods; 2.2 Study 2 methods”. We have combined results of Studies 1 and 2 as follows: “3. Results: 3.1 Study 1 results; 3.2 Study 2 results”. We have changed the title “General discussion” to just “Discussion”. 

Comments 4:

The first paragraph needs references for all what is said in there.

Response 4: Thank you for pointing this out. We have added references to the first paragraph [Line 28-39].

That is, “ Project-based learning is globally popular in education. For example, the project-based learning Works at the Buck Institute for Education, USA, has been providing professional development in science project-based learning to teachers and leaders at both the school and district levels since 1999 (Kingston, 2018). Finland’s new National Curriculum for 2016 emphasizes interdisciplinary and integrated education to dismantle subject barriers and prepare students for future needs through project-based learning (Braskén, Hemmi & Kurtén, 2020). According to the “Compulsory Education Curriculum Plan and Curriculum Standards (2022)” of the Ministry of Education of the People's Republic of China, two basic principles for the compulsory education curriculum were implemented: one was to strengthen curriculum integration and focus on curriculum correlations and the other was to transform the educational approach to highlight practice, advocating for learning through doing, using, and creating.”

Comments 5:

In section 1.1, Hung et al., (2019) cannot be used as a reference for project based teaching because authors talk about project based learning.

Response 5: We apologize for the lack of clarity. We cite this study as a definition of project-based learning, not project-based teaching [Line 53-56; Line 83-85]. That is “Hung et al. (2019) proposed that the core model of project-based learning includes problem-driven instruction, self-directed learning, and small-group learning to structure students’ studies. ” “Compared with the considerable body of literature on project-based learning (Hung et al., 2019; Kokotsaki et al., 2016; Zhang & Ma, 2023), relatively little attention has been paid to project-based teaching.”

Comments 6:

You cannot take out some Buck Institute's standards just by saying "we think" (line 61). You have to prove this.

Response 6: Thank you for pointing this out. We made two adjustments to the Buck Institute's standards, mainly from a practical point, and we have included them in the Introduction [Line 69-82].

First, no “build the culture” standard is included due to implementation challenges. "Build the culture" is a broad and somewhat abstract concept that can be challenging to implement and measure. It requires a long-term commitment and a holistic approach that extends beyond the scope of a single project. While building a positive classroom culture is important, it is not a practice that can be easily isolated and directly applied to project-based teaching.

Second, we combined the “scaffolding student learning” standard with the “engage and coach” standard into a project implementation element. In practice, scaffolding and coaching often go hand in hand. Teachers who scaffold student learning already engage and coach students by providing the necessary support and guidance. Conversely, effective coaching involves understanding when and how to scaffold learning to meet individual student needs. Combining these practices into a single element allows teachers to focus on a unified approach to supporting student learning rather than distinguishing between two closely related activities.

Comments 7:

It is not clear why you are looking to search for a link between cognitive abilities and project-based teaching.

Response 7: Thank you for pointing this out. We have added a paragraph to explain why we need to take a cognitive perspective [Line 124-136].

That is, “Cognitive ability refers to any ability that concerns the processing of mental information (Carroll, 1993). Such cognitive abilities are foundational to complex tasks. For example, studies on problem-solving highlight that spatial and verbal abilities are critical for representing abstract relationships and coordinating multi-step processes (Hegarty & Kozhevnikov, 1999; Newcombe & Shipley, 2015; Wang et al., 2022). As a complex teaching task, project-based teaching may also require cognitive abilities. For example, teachers with higher spatial ability demonstrate superior curriculum integration in STEM fields (Uttal et al., 2013). Meanwhile, teachers with stronger verbal abilities excel in scaffolding student discussions (Hattie, 2012). Crucially, cognitive abilities are not merely “static traits” but malleable skills that can be enhanced through targeted training (Hawes et al., 2022; Rogge et al., 2017; Uttal et al., 2013). By examining teachers’ cognition, this study seeks to uncover actionable pathways for professional development.”

Comments 8:

There is no discussion for any of the studies. Results should be contrasted to those obtained by other authors to know if they are good or new.

Response 8: Thank you for pointing this out. We agree with this comment. Therefore, we have contrasted our study with related studies and included this contrast in the Discussion [Line 514-527].

That is, “In the field of project-based teaching, previous studies on teachers’ project-based learning have explored the influencing factors of gender, implementation experience, and school management (Gomez-Pablos et al., 2017); professional experience and teaching direction (Rogers et al., 2011); attitudes towards project-based learning (Habok & Nagy, 2016); multimedia resources (Ozdamli, 2011); teacher motivation (Lam et al., 2009); professional identity (Tsybulsky & Muchnik-Rozanov, 2019); and more. Unlike these studies, our study aimed to fill the gap in the literature regarding the role of teachers’ cognitive abilities in project-based teaching. It uncovered an actionable pathway for professional development from a cognitive perspective. In terms of cognitive studies, this study provides new evidence for the practical application of the object–spatial–verbal cognitive model. The results of spatial ability alone have a significant predictive effect, indirectly demonstrating the separation and conflict between object, spatial, and verbal abilities. Moreover, the current results demonstrate the particularity and importance of spatial ability in many cognitive abilities.”

Comments 9:

English should be checked. There are phrases that cannot be understood. For example "More importantly, as cognitive ability is an important ability in human, there is no research to explore the role of cognitive abilities on project-based teaching", or "Project-based teaching as another major subject of project-based learning, it also needs to be paid attention to.", or "In teaching class, the bulid of culture is not formed in one day, nor is it the content that must be completed in every class of project-based teaching, so we do not think it is the core content."

Response 9: Thank you for pointing this out. We have modified these sentences.

[Line 43-48]“However, cognitive abilities are foundational to complex tasks (Hegarty & Kozhevnikov, 1999; Newcombe & Shipley, 2015; Wang et al., 2022). As a complex teaching task, project-based teaching may also require cognitive abilities. Currently, no studies have explored the role of cognitive abilities in project-based teaching.”

[Line 40-41] “As an essential part of project-based learning, project-based teaching must be given due consideration. ”

We have deleted “In teaching class, the bulid of culture is not formed in one day, nor is it the content that must be completed in every class of project-based teaching, so we do not think it is the core content.”

Comments 10:

Comments on the Quality of English Language

This text needs serious work. Authors don't present a strong argumentation to know that this work will extend the frontier of knowledge. There seems that there is no need for this research. Additionally, the writing is so poorly executed that it does not allow understanding of the text.

Response 10: Thank you for pointing this out. We have carefully proofread the manuscript, and we have enlisted a professional English editing service to revise it..

Reviewer 6 Report

Comments and Suggestions for Authors

Review Report

This study provides a significant and valuable contribution to the field of education by exploring the relationship between teachers’ cognitive abilities, particularly spatial ability, and their effectiveness in project-based teaching. Unlike prior research that predominantly focuses on students' cognitive skills, this study shifts the lens to educators, addressing an important and underexplored gap in the literature. The mixed-method approach adopted in this research is commendable, as it combines robust quantitative statistical analyses with detailed qualitative insights, offering a holistic perspective on the topic. The findings reinforce the critical role of spatial abilities in the effective implementation of project-based teaching, providing actionable implications for designing teacher training programs and improving curriculum planning. Additionally, the study is firmly anchored in the Object-Spatial-Verbal Cognitive Model, a well-established theoretical framework, and employs validated tools like the Paper Folding Task, enhancing its academic rigor and alignment with the research purpose.

While the measurement tools used in the study demonstrated acceptable reliability (Cronbach’s α ≥ 0.7), they revealed notable limitations in capturing the full spectrum of cognitive abilities relevant to project-based teaching. For instance, the correlation analysis indicated a significant relationship between spatial ability and all elements of project-based teaching, affirming its centrality. However, other cognitive abilities such as creativity, verbal ability, and object detail memory displayed weak or nonsignificant correlations with the teaching constructs. This disparity highlights a possible misalignment between the measurement tasks and the multifaceted demands of project-based teaching. The Remote Association Task, employed to assess creativity, predominantly focuses on linguistic associations and fails to address nonverbal and practical aspects of creativity, such as visualizing and constructing complex projects. Similarly, the Sentence Comprehension Task, while effective in evaluating linguistic comprehension, does not account for critical instructional and communicative skills essential for guiding learners in a project-based environment. Furthermore, the Object Detail Memory Task showed no meaningful association with teaching elements, raising concerns about its relevance to the research objectives. These limitations in measurement tools might have resulted in an underestimation of the role of certain cognitive abilities, thereby influencing the comprehensiveness of the study's findings.

The qualitative component of the study, although providing valuable insights, also underscores a restricted focus on spatial ability, to the exclusion of other cognitive traits that may influence project-based teaching. By centering solely on spatial ability in the case studies, the research misses an opportunity to investigate the interplay of multiple cognitive abilitiessuch as creativity and verbal skillsand their collective impact on teaching efficacy. The qualitative analysis would have benefited from broader inclusion criteria, integrating a wider array of cognitive traits to offer a more nuanced and multidimensional understanding of their role in project-based teaching. Additionally, the reliance on only two teacher case studies limits the generalizability of the findings, as it does not account for variations in teaching styles, educational contexts, or individual teacher characteristics. Incorporating observations from real classroom interactions or expanding the qualitative sample size would have enriched the analysis, contextualized the statistical findings, and provided a more representative depiction of how cognitive abilities manifest in teaching practices.

Author Response

Thank you very much for taking the time to review this manuscript. Please find the detailed responses below and the corresponding revisions in the re-submitted files.

Comments 1:

While the measurement tools used in the study demonstrated acceptable reliability (Cronbach’s α ≥ 0.7), they revealed notable limitations in capturing the full spectrum of cognitive abilities relevant to project-based teaching. For instance, the correlation analysis indicated a significant relationship between spatial ability and all elements of project-based teaching, affirming its centrality. However, other cognitive abilities such as creativity, verbal ability, and object detail memory displayed weak or nonsignificant correlations with the teaching constructs. This disparity highlights a possible misalignment between the measurement tasks and the multifaceted demands of project-based teaching. The Remote Association Task, employed to assess creativity, predominantly focuses on linguistic associations and fails to address nonverbal and practical aspects of creativity, such as visualizing and constructing complex projects. Similarly, the Sentence Comprehension Task, while effective in evaluating linguistic comprehension, does not account for critical instructional and communicative skills essential for guiding learners in a project-based environment. Furthermore, the Object Detail Memory Task showed no meaningful association with teaching elements, raising concerns about its relevance to the research objectives. These limitations in measurement tools might have resulted in an underestimation of the role of certain cognitive abilities, thereby influencing the comprehensiveness of the study's findings.

Response 1: Thank you for pointing this out. We agree with this comment. We have included the limitations of current cognitive tasks in capturing the full spectrum of cognitive abilities in the Discussion [Line 528-544].

That is, “The correlation and regression analyses did not identify any associations between primary school teachers’ creative, object, and verbal abilities and project-based teaching. While creativity enhances student innovation in project-based learning (Ersoy, 2014), remote association tasks may only capture linguistic creativity rather than nonverbal creativity (e.g., visualizing and constructing complex tasks) or pedagogical creativity (e.g., adapting projects to diverse learners). This may be one of the reasons why creativity is not significant. The object–spatial–verbal cognitive model proposes interferences and trade-offs among abilities due to competition for limited executive resources (Blazhenkova & Kozhevnikov, 2009). When spatial abilities are advantageous, object and verbal abilities may not be significant. Our project-based teaching questionnaire focused on static measurement rather than real-time interaction in real classrooms, potentially underestimating the role of object and verbal abilities. In addition, the reason may also lie in the limitations of the current measurement tasks. For example, the sentence comprehension task does not account for critical instructional and communicative skills that are essential for guiding learners in a project-based environment. The object detail memory task also showed no meaningful association with teaching elements. This needs to be further explored in future studies.” 

Comments 2:

The qualitative component of the study, although providing valuable insights, also underscores a restricted focus on spatial ability, to the exclusion of other cognitive traits that may influence project-based teaching. By centering solely on spatial ability in the case studies, the research misses an opportunity to investigate the interplay of multiple cognitive abilities—such as creativity and verbal skills—and their collective impact on teaching efficacy. The qualitative analysis would have benefited from broader inclusion criteria, integrating a wider array of cognitive traits to offer a more nuanced and multidimensional understanding of their role in project-based teaching.

Response 2: Thank you for pointing this out. We agree with this comment, Study 2 should integrate a wider array of cognitive traits to offer a more nuanced and multidimensional understanding of their role in project-based teaching. We regret that Study 2 cannot provide further cases.

The teachers in Study 2 were recruited from the participants in Study 1. After data collection in Study 1, all teachers began to develop project-based teaching plans, but fewer than 10 teachers completed the final plans. Therefore, for Study 2, we selected two teachers from among the Study 1 participants who had different spatial abilities and had completed the final plans for further analysis. Teacher A had good spatial ability (paper folding: 10), while teacher B had less spatial ability (paper folding: 3); however, there were no differences in other aspects (see details in Table 1). For example, they were both math teachers from Beijing, China. There were no differences in their other cognitive abilities, including remote association, sentence comprehension, and object detail memory. However, due to a limited number of cases, we could not find teachers with differences in creativity, verbal, or object ability and no differences in other aspects. In future studies, the sample size needs to be further expanded to find matching cases.

Comments 3:

Additionally, the reliance on only two teacher case studies limits the generalizability of the findings, as it does not account for variations in teaching styles, educational contexts, or individual teacher characteristics. Incorporating observations from real classroom interactions or expanding the qualitative sample size would have enriched the analysis, contextualized the statistical findings, and provided a more representative depiction of how cognitive abilities manifest in teaching practices.

Response 3: Thank you for pointing this out. We agree with this comment. Therefore, we have included this information as part of the limitations [Line 579-591].

That is, “First, the sample size was small. With only 62 primary school teachers involved, the findings may not be broadly generalizable to the larger population of educators. The two teachers in Study 2 probably cannot speak for the entire group, as this sample size does not account for variations in teaching styles, educational contexts, or individual teacher characteristics. A smaller sample size can limit the statistical power of the study, potentially leading to less reliable and less robust results. Future research should aim to include a more extensive and diverse sample of teachers to enhance the generalizability and reliability of the findings. 

Second, the investigation of project-based teaching in this study was mainly based on self-reporting. Incorporating observations from real classroom interactions would have enriched the analysis and provided a more representative depiction in teaching practices. Perhaps based on new technologies such as virtual reality and artificial intelligence, immersive and personalized project-based teaching ability measurement can be achieved. ”

Round 2

Reviewer 4 Report

Comments and Suggestions for Authors

Dear authors:

I consider the improvements made to the article to be very relevant.

Rew

Author Response

Comment 1:

Dear authors:

I consider the improvements made to the article to be very relevant.

Rew

Response : Thank you very much for your positive feedback on the revised manuscript. We truly appreciate your time and effort in reviewing our manuscript.

Reviewer 5 Report

Comments and Suggestions for Authors

This work studies the relationship between cognitive abilities in teachers to their skills to conduct project-oriented teaching. The soul of this study is very relevant, however, The process must be polished yet. 

First paragraph of section 1.1 is mixing project-based learning and project-based teaching concepts and using them interchangeably even if section is about project-based teaching. This happens throughout the entire document as well.

Explanation in lines 67 to 73 is not clear. Saying that something is hard to do is not enough to say that a golden rule (maybe meaning essential standard) has been ignored.

For explanation in lines 73 to 80, theoretical foundation is needed. If this theory is not present, text remains as a personal opinion. 

Text in lines 64 to 80 should be placed in a separated section. It seems that it is an authors' contribution so it has to be explained in detail and must have a theoretical foundation.

Authors should adjust the abstract section to introduce the "regression" instead of leaving the analysis in a "correlation" level. 

Specify the exact number of schools in line 208.

I highly recommend to present methodology of results for study 1 before present those elements for study 2 because the last needs information of the first one. Furthermore, I highly recommend to take out study 2. It would be very interesting to make the analysis using a process to observe the real cognitive abilities (as study 2 suggests), however the study cannot be generalized using a sample size of 2.

"The" is repeated in line 340.

Questions in appendix A are not measuring cognitive abilities. They are measuring teachers' perceptions about their cognitive abilities. So real cognitive abilities cannot be related to project-oriented teaching because they are not directly observables by means of personal perceptions. Given this, authors should consider the Dunning-Kruger effect. You mention this in lines 578 to 583 but you described the study as using real cognitive abilities. 

Do you mean "should be incorporated" instead of "will be incorporated" in line 541?

Conclusion section should be extended with elements that are in other sections. For example, a summary of the research and the results should be presented here. Also, limitations should be presented here. The summary of the contributions of this work should be extended to highlight their importance. 

Author Response

Review 5:

Thank you very much for taking the time to review this manuscript again. Please find the detailed responses below and the corresponding revisions in the re-submitted files.

Comment 1:

First paragraph of section 1.1 is mixing project-based learning and project-based teaching concepts and using them interchangeably even if section is about project-based teaching. This happens throughout the entire document as well.

Response 1: Thank you for pointing this out. We agree with this comment. Therefore, we have removed project-based learning related descriptions and emphasized project-based teaching’s role in first paragraph of 1.1. Since project-based teaching and learning are indeed inseparably linked, we have retained the description of their relationship.

That is “Project-based teaching is essentially another aspect of project-based learning but for different subjects. Generally, project-based learning is a learning process for students, while project-based teaching refers to a process through which teachers design and guide students in implementing project-based learning in class (Grossman et al., 2019). Project-based teaching has no single, precise definition, its advocates generally agree on certain basic characteristics. For example, Hasni et al. (2016) reviewed 48 articles from 2004 to 2014, proposed five main features to define project-based science and technology teaching and learning: there is an authentic scientific question; the students develop a final product; the students are engaged in investigations or design activities; there is collaboration among students, teachers and others in the community; and learning technologies are used. Grossman et al., (2019) proposed basic characteristics of project-based teaching including: giving students opportunities to study a challenging problem, engage in sustained inquiry, find answers to authentic questions, help choose the project, reflect on the process, critique and revise the work, and create a public product. More importantly, the Buck Institute for Education proposed more practical seven gold standards for project-based teaching, including design and plan, align to standard, build the culture, manage active, scaffold student learning, assess student learning, engage and coach (Larmer et al., 2015).”

Second, we have deleted the project-based learning related descriptions in other aspects. For example, the first sentence of previous Abstract is “Project-based learning has become popular worldwide in recent years. However, previous studies focused more on students’ project-based learning and less on teachers. And the role of teachers’ cognitive abilities on project-based teaching is still unknown. ” We have changed as “Cognitive abilities are foundational to complex tasks, which may be also important in complex project-based teaching. However, the role of teachers’ cognitive abilities on project-based teaching is still unknown. ”

Comment 2:

Explanation in lines 67 to 73 is not clear. Saying that something is hard to do is not enough to say that a golden rule (maybe meaning essential standard) has been ignored.

For explanation in lines 73 to 80, theoretical foundation is needed. If this theory is not present, text remains as a personal opinion.

Text in lines 64 to 80 should be placed in a separated section. It seems that it is an authors' contribution so it has to be explained in detail and must have a theoretical foundation.

Response 2: We apologize for the unclear description in previous version.

(1) Text in line 64 to 80 provides the theoretical foundation for the measurement of project-based teaching in this study. Therefore, we have relocated this section to “1.3 Current study”.

(2) We have added the more evidences about no “build the culture” standard is included (previous explanation in lines 67 to 73).

That is “First, no “build the culture” standard is included due to implementation challenges. This standard ask teachers to explicitly and implicitly promote student independence and growth, open-ended inquiry, team spirit, and attention to quality (Larmer et al., 2015). This shows that the standard itself has a characters of implicitly. Moreover, the different school culture background, student diversity, and uneven teaching resource distribution can make it challenging to implement the cultural related standards. For example, a qualitative study showed that elementary teachers may not be teaching in a culturally responsive way due to a lack of classroom models (Hawkins, 2021).”

  • We have added the more evidences about combined the “scaffolding student learning” standard with the “engage and coach” standard into a project implementation element(explanation in lines 73 to 80).

That is “Second, we combined the “scaffolding student learning” standard with the “engage and coach” standard into a project implementation element. The “scaffolding student learning” emphasizes provide a variety of strategies to support students, while "engage and coach" focuses on the dynamic interaction of students and teachers in the process (Larmer et al., 2015). Vygotsky's (1978) “zone of proximal development” theory pointed out that learning should be realized through scaffolding and interaction under the guidance of teachers. With the combination, teachers can both provide instrumental support and dynamically adjust instruction through observation and feedback. In practice, scaffolding and coaching often go hand in hand. Teachers who scaffold student learning already engage and coach students by providing the necessary support and guidance.”

Comment 3:

Authors should adjust the abstract section to introduce the "regression" instead of leaving the analysis in a "correlation" level.

Response 3: Thank you for pointing this out. We agree with this comment. We have modified Abstract to introduce regression in Study 1.

That is “In Study 1, a quantitative regression analysis was conducted with 62 primary school teachers. They completed the project-based teaching questionnaire and performed four cognitive tasks: remote association (creativity), object detail memory (object detail processing ability), paper folding (spatial ability), and sentence comprehension (verbal ability). Regression analysis revealed that spatial ability significantly predicted teacher’s project-based teaching ability, even after controlling for age, gender, teaching experience and project-based teaching experience. ”

Comment 4:

Specify the exact number of schools in line 208.

Response 4: Thank you for pointing this out. We have added the exact number of schools. That is “This study included 62 primary school teachers from 42 schools in Beijing and Hebei, China. ”

Comment 5:

I highly recommend to present methodology of results for study 1 before present those elements for study 2 because the last needs information of the first one. Furthermore, I highly recommend to take out study 2. It would be very interesting to make the analysis using a process to observe the real cognitive abilities (as study 2 suggests), however the study cannot be generalized using a sample size of 2.

Response 5: Thank you for pointing this out.

(1) We have combined the methods and results of Study 1 as one section, that is “2. Study 1; 2.1 Methods; 2.2 Results”. Similarly, we have combined the methods and results of Study 1 as one section, that is “3. Study 2; 3.1 Methods; 3.2 Results”.

(2) Regarding your suggestion to delete Study 2, we fully understand your concern that the small sample size (only 2 participants) may affect the generalization of the results. However, we respectfully request that Study 2 be retained for the following reasons: 

First, the design orientation of Study 2 is an exploratory case study. The core objective is not to generalize statistics, but to explore the potential mechanisms of application of cognitive ability in educational practice. Such case study is of great value in the early stage of theory construction or method validation (Flyvbjerg, 2006).

Second, Study 2 have unique scientific contribution in current study. Through two typical cases, we verified the operation process and sensitivity of the spatial ability in project-based teaching, which laid the foundation for the follow-up large-sample research. And the consistency of the qualitative data with the quantitative results enhances the credibility of the overall conclusion (Creswell & Clark, 2017). 

We have added these reasons in the “3. Study 2”.

Comment 6:

"The" is repeated in line 340.

Response 6: Thank you for pointing this out. We are very sorry for this error, we have corrected this typo and carefully proofread the manuscript.

Comment 7:

Questions in appendix A are not measuring cognitive abilities. They are measuring teachers' perceptions about their cognitive abilities. So real cognitive abilities cannot be related to project-oriented teaching because they are not directly observables by means of personal perceptions. Given this, authors should consider the Dunning-Kruger effect. You mention this in lines 578 to 583 but you described the study as using real cognitive abilities.

Response 7: (1) We are sorry that our description is not clear enough and may have caused misunderstanding. Cognitive abilities were measured via validated performance tasks (e.g., paper folding task for spatial ability), avoiding self-report biases (Dunning-Kruger effect). The project-based teaching questionnaire in Appendix A is evaluated self-perceived instructional skills, distinct from cognitive tasks. 

(2) We also agree with reviewer that the project-based teaching questionnaire does have the problem of measuring teachers’ perceptions rather than ability. We have added the discuss of Dunning-Kruger effect in “3.Study 2”.

That is “However, relying on self-reported measures for project-based teaching may introduce bias. For example, the Dunning-Kruger effect, a cognitive bias wherein individuals with low ability in a domain overestimate their competence, can significantly impact self-report questionnaires (Kruger & Dunning, 1999). This case analysis involves in-depth examination of spatial ability role in project-based teaching, providing a more objective and detailed understanding to avoid self-report biases. ”

Comment 8:

Do you mean "should be incorporated" instead of "will be incorporated" in line 541?

Response 8: Thank you for pointing this out. We have changed this sentences.

Comment 9:

Conclusion section should be extended with elements that are in other sections. For example, a summary of the research and the results should be presented here. Also, limitations should be presented here. The summary of the contributions of this work should be extended to highlight their importance.

Response 9: Thank you for pointing this out. We have added the summary of research, result,contributions and limitations in “5.Conclusions”.

That is “Through a comprehensive approach involving both quantitative (Study 1) and qualitative (Study 2) analyses, this study revealed spatial ability could significantly predicted elementary school teachers’ project-based teaching ability. Specifically, Study 1 found that 62 primary school teacher’ spatial ability, rather than creativity, object detail processing ability and verbal ability, significantly predicted teachers’ project-based teaching ability, even after controlling for age, gender, teaching experience and project-based teaching experience. Study 2 found that the higher spatial ability teacher used more schemata, abstract concepts, better overall plan, more precise key points in project-based teaching plan than teacher with lower spatial ability. This study not only highlights the importance of spatial ability in project-based teaching but also uncovered an actionable pathway for professional development from a cognitive perspective. And the mixed method of qualitative and quantitative analysis enhances the credibility of the overall conclusion. This study has some limitations. For example, the sample size was small; the cognitive measurement tools do not well integrate with educational contexts and so on. Future studies should aim to validate these findings with more extensive and diverse samples and explore the long-term impact of spatial ability on teaching practices. ”

Reviewer 6 Report

Comments and Suggestions for Authors

I sincerely commend your efforts in revising the manuscript and thoughtfully addressing the reviewers’ comments. Your careful revisions reflect a strong commitment to enhancing the clarity, rigor, and overall scholarly contribution of your study.

That being said, while the revised manuscript appropriately acknowledges the limitations of the measurement tools, I would like to draw attention to an important issue. Merely stating these limitations does not sufficiently address the methodological constraints or their potential impact on the study’s findings. Future researchers who may reference this work would greatly benefit from a more comprehensive discussion on how these limitations influenced the results and, crucially, how subsequent research might overcome these challenges.

For instance, while the manuscript notes that the cognitive tasks employed may not fully encapsulate the complexity of project-based teaching, it would be valuable to offer specific recommendations on how future studies might develop or refine measurement instruments to better capture these constructs. Similarly, while the restricted qualitative sample size is acknowledged, a more detailed examination of the potential biases and interpretative limitations arising from this constraint would strengthen the study’s overall validity and applicability. Providing deeper insights into how future research can expand upon and refine these findings would substantially enhance the manuscript’s scholarly impact.

I encourage the authors to not only recognize the study’s limitations but also to critically engage with their broader implications for future research. A more reflective and forward-looking discussion would provide a stronger foundation for subsequent investigations aiming to validate or extend these findings.

Overall, I appreciate the authors’ thoughtful revisions and their engagement with the reviewers’ feedback. I believe that by incorporating these additional considerations, the manuscript will make an even more significant contribution to the field.

Author Response

Review 6:

Thank you very much for taking the time to review this manuscript. Please find the detailed responses below and the corresponding revisions in the re-submitted files.

Comment 1:

That being said, while the revised manuscript appropriately acknowledges the limitations of the measurement tools, I would like to draw attention to an important issue. Merely stating these limitations does not sufficiently address the methodological constraints or their potential impact on the study’s findings. Future researchers who may reference this work would greatly benefit from a more comprehensive discussion on how these limitations influenced the results and, crucially, how subsequent research might overcome these challenges.

For instance, while the manuscript notes that the cognitive tasks employed may not fully encapsulate the complexity of project-based teaching, it would be valuable to offer specific recommendations on how future studies might develop or refine measurement instruments to better capture these constructs. Similarly, while the restricted qualitative sample size is acknowledged, a more detailed examination of the potential biases and interpretative limitations arising from this constraint would strengthen the study’s overall validity and applicability. Providing deeper insights into how future research can expand upon and refine these findings would substantially enhance the manuscript’s scholarly impact.

I encourage the authors to not only recognize the study’s limitations but also to critically engage with their broader implications for future research. A more reflective and forward-looking discussion would provide a stronger foundation for subsequent investigations aiming to validate or extend these findings.

Overall, I appreciate the authors’ thoughtful revisions and their engagement with the reviewers’ feedback. I believe that by incorporating these additional considerations, the manuscript will make an even more significant contribution to the field.

Response 1: We appreciate the reviewer’s insightful feedback. We have revised the “4.4 Limitations and Future Suggestions” to provide a more reflective and forward-looking discussion.

That is “First, the small sample size poses limitations to the generalizability of the findings. With only 62 primary school teachers participating in Study 1, the quantitative results may not fully represent broader regions, schools and so on. A smaller sample size can limit the statistical power of correlation and regression results, potentially leading to less reliable and less robust results. Researchers could divide the population into strata based on key characteristics (e.g., regions, age, teaching subjects) and using large-scale stratified sampling. More critically, the qualitative case study (Study 2) included only two teachers, which does not account for variations in teaching styles, educational contexts, or individual teacher characteristics. Small qualitative samples also risk premature theoretical saturation, where emergent themes may reflect idiosyncratic experiences rather than generalizable patterns (Creswell & Poth, 2018). For example, the contrasting cases in Study 2 may overemphasize spatial ability’s role while neglecting other contextual factors (e.g., school resources, student levels) that influence project-based teaching. To address these limitations, in addition to expanding the sample size, future studies could using multiple data sources (e.g., interviews, observations, student feedback) to provide a more comprehensive understanding. Moreover, a coding scheme could development to help researchers systematically analyze a large amount of data.

Second, there are some limitations of the measurement tools. In the measurement of project-based teaching, it was mainly based on self-reporting (Study1) and text analysis (Study 2). The self-reported measures may introduce bias, such as Dunning-Kruger effect and so on (Kruger & Dunning, 1999). The text analysis may struggle with language characters, such as slang, jargon, or domain-specific language (Turney & Pantel, 2010). Incorporating observations from real classroom interactions would have enriched the analysis and provided a more representative depiction of teaching practices. Perhaps based on new technologies such as virtual reality and artificial intelligence, immersive and personalized project-based teaching ability measurement can be achieved in future studies. In the measurement of cognitive abilities, these are are measures of domain general abilities, but do not well integrate with educational contexts. For example, remote association tasks may only capture linguistic creativity rather than nonverbal creativity (e.g., visualizing and constructing complex tasks) or pedagogical creativity (e.g., adapting projects to diverse learners). The sentence comprehension task does not account for critical instructional and communicative skills that are essential for guiding learners in a project-based environment. The object detail memory task also showed no meaningful association with teaching elements. This may be an important reason for the lack of associations between primary school teachers’ creative, object, and verbal abilities and project-based teaching. Future studies should use a variety of tasks that capture different aspects of cognitive abilities, such as mathematical creativity, speech intelligibility, educational object details memory and so on. The future studies also could combine cognitive tests with classroom observations and AI-driven analytic to development interactive and implicitly measurement tools. For example, mathematical teachers can demonstrate operations such as rotation, symmetry, and translation of graphs in class. AI platform can record the reaction time of these operations in real time, and automatically analysis the data related to spatial ability.”

Round 3

Reviewer 5 Report

Comments and Suggestions for Authors

Thank you to address my comments in this version. I think the text is more clear this time. The only thing that I still have in mind is study 2. I think it cannot be presented as it is and it should be presented (as you say) as a case study or as an example of what was observed in study 1. 

Author Response

Comment 1:

Thank you to address my comments in this version. I think the text is more clear this time. The only thing that I still have in mind is study 2. I think it cannot be presented as it is and it should be presented (as you say) as a case study or as an example of what was observed in study 1.

Response 1: Thank you very much for your positive feedback on the revised manuscript. We truly appreciate your time and effort in reviewing our manuscript. We have revised Study 2 based on this comment to make it more consistent with the presentation of case study [Line 415-488].

That is “3.2 Results

3.2.1 Teacher A: higher spatial ability

Teacher A’s project-based teaching plan was ‘park route planning’, which required students to provide personalized tour route services to address the different needs of tourists and effectively improve their satisfaction.

In the textbook knowledge analysis, she drew a knowledge logical structure diagram. In the diagram, the relationships between knowledge points were expressed through schemata. Schematic representations (drawn picture that can represent quantitative relationships between mathematical elements) are related to the accuracy of word problem solving, as well as spatial ability (Boonen et al., 2013; Boonen et al., 2014; Hegarty & Kozhevnikov, 1999). For teachers, the use of a schema in a knowledge analysis may explain the association of project-based teaching and spatial ability.

In the project-based learning design, teacher A designed a more abstract and general phase of the project. For example, the phases were as follows:

“1: Project release -- clarification of issues; 2: Project exploration - problem solving; and 3: Project optimization -- problem reflection.”

Perhaps for the design of project-based learning, more abstract and general concepts is needed to facilitate its transfer to the design of other projects. At the same time, spatial ability is also a more general, more abstract form of picture expression (Hegarty, 2010). The role of spatial ability is reflected in the abstract of design phase of project-based teaching, rather than the concrete design content. This may also be the reason why the relationship of spatial ability and project-based teaching design ability was not found in Study 1. The evaluation of design ability in Study 1 did not focus on abstractness.

In the implementation procedure, teacher A spent less class hours on the project introduction (1 class) than on project design (2 classes). This showed that teacher A was more better planning of teaching implementation. This ability may be related to spatial ability because the latter is the ability to comprehend the overall abstract relationship, which contributes to overall planning ability (Gonçalves & Ferreira, 2015).

For assessment plans, teacher A’s assessment was closer to the essence of mathematics. For example, the assessment was as follows:

“Excellent: according to the needs of specific groups, the development of special tour routes are in line with the standard, and the route is scientific and reasonable. Good: according to the needs of specific groups of people, the design of special tour lines was implemented, but there is a slight gap with the standard. Qualified: able to design characteristic tour routes, but the design is not reasonable.”

This shows the teacher's understanding of the nature of the mathematics, which in turn affects the depth of the assessment criteria. At the same time, the understanding of mathematics is closely related to spatial ability (Boonen et al., 2014; Hegarty, & Kozhevnikov, 1999; Hawes, et al., 2022). This also explains why the Study 1 did not find a relationship between spatial ability and project-based teaching assessment ability, possibly because this association is closely related to the specific subject.

3.2.2 Teacher B: lower spatial ability

Teacher B’s plan was ‘when art meets mathematics’, which required that to learn to appreciate Escher’s dense-tiled layout artworks, students need to use periodic changes and graphical transformations in mathematics to design and create their own dense-tiled layout artworks.

In the textbook knowledge analysis, teacher B used the pure text description method. For teachers, the use of a pure text description  in a knowledge analysis related with verbal ability. According the interferences and trade-offs among abilities in object–spatial–verbal cognitive model (Blazhenkova & Kozhevnikov, 2009), the advantage of verbal ability may limit the use of spatial ability. Therefore, when Teacher B shows greater preference for verbal ability, his spatial ability may be weaker.

In the project-based learning design, teacher B’s phase was more concrete. For example, the phases of were as follows:

“1: Art appreciation; 2: Concept understanding; 3: Art processing; and 4: Creative design.”

Compared with teacher A, his design of project-based learning focus on specific project. These concrete and specific thinking may not be facilitate spatial ability.

In the implementation procedure, teacher B spent more class hours on the project introduction (2 classes) than on the project design (1 class). It shows that she does not have a good overall plan of the implementation of project-based learning. And this also explains the lack of overall representation ability of spatial ability.

For assessment plans, teacher B’s assessment was not related with the essence of mathematics. For example, teacher B’s assessment was as follows:

“Excellent: understood the design principle of dense-tiled layout, completed dense-tiled layout art work, which was beautifully designed. Share design ideas, creative processes, and challenges. Good: understood the design principle of dense- tiled layout, basically completed a dense-tiled layout art work, with a certain artistic processing. Share design ideas. Qualified: Understood the design principle of dense-tiled layout, and basically completed a dense-tiled layout art work.”

Obviously, teacher B’s assessment was not closely related to mathematical ability but was based on art and sharing. This may be related to his lack of spatial ability.”

Reviewer 6 Report

Comments and Suggestions for Authors

I sincerely appreciate the time and effort you have dedicated to revising your manuscript in response to the reviewer’s comments. Your careful consideration of the feedback and thoughtful revisions have significantly strengthened the clarity, methodological rigor, and scholarly contribution of your work.

I am particularly pleased to see how you have expanded the discussion on methodological limitations, provided a more comprehensive reflection on their potential impact, and offered constructive suggestions for future research. These improvements enhance the robustness and applicability of your findings, making your study a more valuable contribution to the field.

Your commitment to refining the manuscript and addressing the reviewer’s concerns is commendable, and I appreciate your responsiveness and scholarly engagement throughout the revision process. I believe that these enhancements will further enrich academic discourse in this area.

Thank you for your diligent efforts, and I look forward to seeing the continued impact of your research.

Author Response

Thank you very much for your positive feedback on the revised manuscript. We truly appreciate your time and effort in reviewing our manuscript. We will continue the relevant research with your encouragement.